# Spectral-Informed Neural Networks Outperform Spectral Methods in High-dimensional PDEs

**Tianchi Yu** [1 2]  **Ivan Oseledets** [1 3]

## Abstract

For low-dimensional problems ($d \leq 3$), spectral methods can achieve exceptionally high accuracy. For middle-dimensional problems ($4 \leq d \lesssim 10$), spectral methods remain feasible through specific techniques such as sparse grids or hyperbolic cross. However, for high-dimensional problems ($d \gg 10$), spectral methods suffer frome the curse of dimensionality. Physics-informed neural networks (PINNs) have emerged as a promising approach to overcome this challenge, offering scalability to high dimensions, but often suffer from limited accuracy and efficiency. Recently proposed spectral-informed neural networks (SINNs) combine spectral methods with PINNs, operating directly in the spectral domain to avoid spatial derivative computations and to reduce memory consumption. In this work, we introduce Modified SINNs, which integrate coefficient decay scaling and basis embeddings motivated by harmonic analysis to enhance accuracy in high-dimensional problems and enable accurate approximation of unknown spectral coefficients. Numerical experiments on steady and time-dependent partial differential equations demonstrate that Modified SINNs outperform sparse grid spectral methods on middle-dimensional problems with incomplete spectral information and achieve superior accuracy compared to PINNs on high-dimensional problems.

## 1. Introduction

Partial differential equations (PDEs) are the most widely used tools for solving physical and engineering problems (Karniadakis & Sherwin, 2005). Many applications (e.g., in financial engineering, quantum mechanics) lead to high-dimensional problems (Peherstorfer et al., 2015a). Numerical methods, such as spectral methods, are the backbone for computing numerical solutions of PDEs. Conventional numerical methods can compute accurate solutions for low-dimensional PDEs ($d \leq 3$). For middle-dimensional PDEs ($4 \leq d \lesssim 10$), the equations can be tackled using techniques, for example, combining sparse grids (Bungartz & Griebel, 2004) with finite element methods (Peherstorfer et al., 2015b), combining compressed sensing (Candès & Wakin, 2008) with spectral methods (Brugiapaglia et al., 2018), and combining tensor trains (Oseledets, 2011) with backward stochastic differential equations (Chertkov & Oseledets, 2021; Richter et al., 2024). However, for high-dimensional PDEs ($d \gg 10$), conventional numerical methods suffer from the curse of dimensionality (CoD), *i.e.*, the computational cost grows exponentially with the dimension.

Deep learning methods have demonstrated an impressive ability to approximate high-dimensional and highly complex functions, as evidenced by their remarkable success in computer vision and natural language processing (Le-Cun et al., 2015). In the context of numerical PDEs, deep learning methods also exhibit strong potential for tackling high-dimensional problems, such as physics-informed neural networks (PINNs) (Raissi et al., 2019) and the deep Ritz method (DRM) (Weinan & Yu, 2018). For PINNs, recent studies (Han et al., 2018; Shi et al., 2024) have increased the maximum achievable dimensionality to one million dimensions. Some current works integrate conventional numerical methods into PINNs to enhance their performance. For example, CANPINN (Chiu et al., 2022) incorporates finite difference methods, DFVM (Cen & Zou, 2024) adopts finite volume methods, and SINN (Yu et al., 2025) employs spectral methods.

In this work, we propose using Spectral-Informed Neural Networks (SINNs) to solve high-dimensional PDEs. By incorporating prior knowledge from harmonic analysis—specifically including coefficient decay and basis-dependent embeddings—our Modified SINNs improve stability and accuracy. Furthermore, by leveraging this prior knowledge, the modification enables SINNs not only to compute known spectral coefficients efficiently, but also to approximate missing or uncomputed coefficients.

---

[1]AXXX [2]Applied AI Institute [3]INM. Correspondence to: Tianchi Yu <807966083@qq.com>.

*Proceedings of the 43rd International Conference on Machine Learning*, Seoul, South Korea. PMLR 306, 2026. Copyright 2026 by the author(s).

Our specific contributions can be summarized as follows:

- We propose *Modified Spectral-Informed Neural Networks* (Modified SINNs), which incorporate prior knowledge from harmonic analysis into the original SINN framework, leading to improved stability and accuracy.

- We reveal that traditional sparse grid spectral methods are still limited by the curse of dimensionality in high-dimensional problems, while SINNs provide an effective framework for solving high-dimensional PDEs. Furthermore, by approximating missing or uncomputed spectral coefficients, SINNs can improve the accuracy of spectral methods.

- Through extensive numerical experiments, we demonstrate that Modified SINNs outperform sparse grid spectral methods on middle-dimensional problems when some coefficients are missing and achieve superior accuracy on high-dimensional problems.

The paper is structured as follows. Section 2 reviews spectral methods, physics-informed neural networks, and spectral-informed neural networks. Section 3 presents the proposed Modified SINNs. Section 4 reports numerical experiments on coefficient prediction, comparisons with sparse grid spectral methods, and comparisons with PINNs and DRMs. Finally, Section 5 concludes the paper and discusses future directions.

## 2. Preliminary

Consider a general PDE defined on a spatial domain $\Omega \subset \mathbb{R}^d$:

$$\begin{aligned} \mathcal{N}[u](\boldsymbol{x}, t) &= f(\boldsymbol{x}, t), & (\boldsymbol{x}, t) \in \Omega \times [0, T], \\ u(\boldsymbol{x}, t) &= g(\boldsymbol{x}, t), & (\boldsymbol{x}, t) \in \partial\Omega \times [0, T], \\ u(\boldsymbol{x}, 0) &= h(\boldsymbol{x}), & \boldsymbol{x} \in \Omega. \end{aligned} \quad (1)$$

Here, $\mathcal{N}$ is a differential operator including spatial and temporal derivatives, $f$ is a forcing term, and $u$ is the solution. In Equation (1), the first equation is called the governing equation, the second equation is called the boundary condition, and the third equation is called the initial condition. In the following, we assume that Equation (1) is well-posed and that $u : \mathbb{R}^d \to \mathbb{R}$.

### 2.1. Spectral Methods

Spectral methods are a class of numerical methods for solving differential equations by representing the solution as a global expansion in terms of spectral bases. Unlike local discretization schemes, spectral methods achieve high accuracy by exploiting the global regularity of the solution (Trefethen, 2000; Boyd, 2001). When the target function is

sufficiently smooth, the convergence rate of spectral approximations is often exponential, a property known as spectral accuracy (Canuto et al., 2007). Due to their accuracy and efficiency for smooth solutions, spectral methods have been widely applied in fluid dynamics (Canuto et al., 2007), quantum mechanics (Deloff, 2007), electromagnetism (Wang & Jiang, 2024), and beyond.

In the spectral method, the solution $u(\boldsymbol{x}, t)$ in Equation (1) is approximated by an expansion.

$$u(\boldsymbol{x}, t) = \sum_{\boldsymbol{k} \in \mathcal{K}} \hat{u}_{\boldsymbol{k}}(t)\phi_{\boldsymbol{k}}(\boldsymbol{x}), \quad (2)$$

In practice, the truncated expansion is used:

$$u_N(\boldsymbol{x}, t) = \sum_{\boldsymbol{k} \in \mathcal{K}_N} \hat{u}_{\boldsymbol{k}}(t)\phi_{\boldsymbol{k}}(\boldsymbol{x}), \quad (3)$$

where $\{\phi_{\boldsymbol{k}}\}$ are global basis functions[1] (e.g., Fourier or Chebyshev), $\hat{u}_{\boldsymbol{k}}$ are expansion coefficients, and $\mathcal{K} = \{\boldsymbol{k} : \|\boldsymbol{k}\|_\infty < +\infty\}$ is the index set and $\mathcal{K}_N$ is a truncated set of $\mathcal{K}$. The coefficients $\hat{u}_{\boldsymbol{k}}$ are determined by Galerkin methods (Equation (21)) or collocation methods (Equation (22))(See Appendix B for details of spectral methods.).

However, spectral accuracy is limited for complex geometries, high dimensionality, or non-smooth solutions. To enhance the applicability of spectral methods under these challenging cases, various extensions have been proposed, including spectral element methods (Karniadakis & Sherwin, 2005), fictitious domain spectral methods (Gu & Shen, 2021), and sparse grid spectral methods (Shen & Wang, 2010).

The sparse grid spectral method (SGSM) is a variant of conventional spectral methods that mitigates the curse of dimensionality by employing a sparser grid and computing fewer spectral coefficients. In some problems, this approach can achieve accuracy comparable to that of full-grid spectral methods, but with significantly lower memory usage and computational cost (Shen & Yu, 2012; Temlyakov, 2014; Wang & Brugiapaglia, 2024). However, the memory usage and computational cost still increase exponentially with dimensionality, which limits its applicability to high-dimensional problems ($d \gg 10$). We point out this phenomenon in Appendix D.

In this paper, the algorithms and schemes of SGSM developed in (Shen & Yu, 2010; 2012) are adopted as the baseline in the experiments. These methods employ hyperbolic cross[2] index sets to construct sparse grids and introduce efficient, well-established algorithms for solving PDEs within the SGSM framework.

---

[1] See Appendix A for details of other spectral bases.
[2] See Appendix C for details of hyperbolic cross.

## 2.2. Physics-informed Neural Networks

Physics-informed neural networks (PINNs) have emerged as an effective framework for solving problems constrained by physical laws, especially partial differential equations (PDEs). Unlike traditional data-driven neural networks, PINNs embed governing equations into the loss function, thereby enabling training without labeled data. In PINN frameworks, the solution $u(\boldsymbol{x}, t)$ in Equation (1) is approximated by a neural network $u^\theta(\boldsymbol{x}, t)$ parameterized by $\theta$. The network is trained to minimize a composite loss function that enforces physical constraints:

$$\mathcal{L}(\theta) = \lambda_r \mathcal{L}_r(\theta) + \lambda_b \mathcal{L}_b(\theta) + \lambda_i \mathcal{L}_i(\theta) \qquad (4)$$

where $\lambda_r, \lambda_b, \lambda_i$ are nonnegative weights. Let $N_r, N_b, N_i$ denote the cardinality of the corresponding training sets; the individual terms in Equation (4) are

$$\mathcal{L}_r(\theta) = \frac{1}{N_r} \sum_{j=1}^{N_r} \left| \mathcal{N}[u^\theta](\boldsymbol{x}_j^r, t_j) - f(\boldsymbol{x}_j^r, t_j) \right|^2, \quad (5)$$

$$\mathcal{L}_b(\theta) = \frac{1}{N_b} \sum_{j=1}^{N_b} \left| u^\theta(\boldsymbol{x}_j^b, t_j) - g(\boldsymbol{x}_j^b, t_j) \right|^2, \qquad (6)$$

$$\mathcal{L}_i(\theta) = \frac{1}{N_i} \sum_{j=1}^{N_i} \left| u^\theta(\boldsymbol{x}_j^i, 0) - h(\boldsymbol{x}_j^i) \right|^2, \qquad (7)$$

which enforce the PDE, boundary conditions, and initial conditions, respectively. By minimizing $\mathcal{L}(\theta)$, the network learns an approximation $u^\theta$ that simultaneously satisfies the governing PDE, boundary conditions, and initial conditions. Automatic differentiation provides efficient evaluation of derivatives in $\mathcal{N}[u^\theta]$, eliminating the need for explicit discretization of differential operators. Thus, unlike conventional numerical schemes, PINNs are particularly attractive for solving high-dimensional problems.

## 2.3. Spectral-Informed Neural Networks

Inspired by PINNs, Spectral-Informed Neural Networks (SINNs) (Yu et al., 2025) are proposed to integrate spectral methods (Section 2.1) into neural networks. SINNs accelerate training by eliminating the computation of spatial derivatives. Moreover, due to the spectral convergence of spectral bases, SINNs obtain better accuracy than PINNs. SINNs reconstruct the loss function (Equation (4)) using the collocation formulation (Equation (22)). Suppose $N_p$ is the number of points used for the discrete forward transform, $\mathcal{T}$ is the set of timestamps sampled randomly from $[0, T]$ (Here, $\mathcal{T}_f$ is the set for the residual loss and $\mathcal{T}_b$ is the set for the boundary loss), and $\mathcal{K}_N$ is the index set (Here, $\mathcal{K}_{N_f}$ is the set for the residual loss, $\mathcal{K}_{N_b}$ is the set for the boundary loss and $\mathcal{K}_{N_i}$ is the set for the initial loss); then the loss

function of SINNs is defined as follows:

$$\hat{\mathcal{L}}(\theta) = \lambda_f \hat{\mathcal{L}}_f(\theta) + \lambda_b \hat{\mathcal{L}}_b(\theta) + \lambda_i \hat{\mathcal{L}}_i(\theta) \qquad (8)$$

where

$$\hat{\mathcal{L}}_f(\theta) = \frac{1}{|\mathcal{T}_f| N_p} \sum_{t \in \mathcal{T}_f} \sum_{j=1}^{N_p} \left| \mathcal{N} \left[ \sum_{\boldsymbol{k} \in \mathcal{K}_{N_f}} \hat{u}_{\boldsymbol{k}}^\theta(t) \phi_{\boldsymbol{k}}(\boldsymbol{x}_j) \right] \right.$$
$$\left. - \sum_{\boldsymbol{k} \in \mathcal{K}_{N_f}} \hat{f}_{\boldsymbol{k}}(t) \phi_{\boldsymbol{k}}(\boldsymbol{x}_j) \right|^2 . \qquad (9)$$

$$\hat{\mathcal{L}}_b(\theta) = \frac{1}{|\mathcal{T}_b| N_p} \sum_{t \in \mathcal{T}_b} \sum_{j=1}^{N_p} \sum_{\boldsymbol{k} \in \mathcal{K}_{N_b}} \left| \hat{u}_{\boldsymbol{k}}^\theta(t) \phi_{\boldsymbol{k}}(\boldsymbol{x}_j) - \hat{g}_{\boldsymbol{k}}(t) \phi_{\boldsymbol{k}}(\boldsymbol{x}_j) \right|^2,$$
$$= \frac{1}{|\mathcal{T}_b|} \sum_{t \in \mathcal{T}_b} \sum_{\boldsymbol{k} \in \mathcal{K}_{N_b}} \left| \hat{u}_{\boldsymbol{k}}^\theta(t) - \hat{g}_{\boldsymbol{k}}(t) \right|^2 . \qquad (10)$$

$$\hat{\mathcal{L}}_i(\theta) = \frac{1}{N_p} \sum_{j=1}^{N_p} \sum_{\boldsymbol{k} \in \mathcal{K}_{N_i}} \left| \hat{u}_{\boldsymbol{k}}^\theta(0) \phi_{\boldsymbol{k}}(\boldsymbol{x}_j) - \hat{h}_{\boldsymbol{k}} \phi_{\boldsymbol{k}}(\boldsymbol{x}_j) \right|^2,$$
$$= \sum_{\boldsymbol{k} \in \mathcal{K}_{N_i}} \left| \hat{u}_{\boldsymbol{k}}^\theta(0) - \hat{h}_{\boldsymbol{k}} \right|^2 . \qquad (11)$$

The second equality in each of Equations (10) and (11) holds due to the orthogonality of $\phi$. Whether the loss terms include the normalization factor $\frac{1}{|\mathcal{K}_N|}$ (where $N \in \{N_f, N_b, N_i, N_p\}$) depends on the implementation of the discrete transform. By minimizing $\hat{\mathcal{L}}(\theta)$, the network learns an approximation $\hat{u}^\theta$ of the coefficients $\hat{u}$. If the solution $u(\boldsymbol{x}, t)$ is required in the physical domain, one can use Equation (3) to compute its value at any point $(\boldsymbol{x}, t) \in \Omega \times [0, T]$. Furthermore, if the spectral basis inherently satisfies the boundary condition (e.g., periodic for the Fourier basis, homogeneous Dirichlet for the sinusoidal basis, or homogeneous Neumann for the cosine basis), then $\hat{\mathcal{L}}_b(\theta)$ is removed from Equation (8). Herein, one of the impressive features is that SINNs can satisfy the boundary condition strictly for some PDEs.

Compared to PINNs, in SINNs, the input is the frequency $\boldsymbol{k}$ and the temporally sampled points $t$, and the output is the coefficients $\hat{u}^\theta(\boldsymbol{k}, t)$[3] in the spectral domain. Similar to PINNs, for each iteration in SINNs, the input sets $\mathcal{K}_{N_f}, \mathcal{K}_{N_b}, \mathcal{K}_{N_i}$ are randomly sampled from $\mathcal{K}_{N_p}$ which is the index set generated by $N_p$ discrete points. However, when $\mathcal{N}$ is nonlinear or the differential operator of $\phi$ is not diagonal, SINNs become unstable if the input sets are not large enough. Herein, this paper proposes Modified SINNs that integrate prior knowledge from harmonic analysis to enhance stability and accuracy.

## 3. Modified SINNs

In this section, we introduce prior knowledge into SINNs and propose Modified SINNs. In particular, Modified

---

[3]We use the notation $\hat{u}^\theta(\boldsymbol{k}, t)$ rather than $\hat{u}_{\boldsymbol{k}}^\theta(t)$ to highlight the input of $\hat{u}^\theta$ is $(\boldsymbol{k}, t)$

SINNs preserve the input–output form $((\boldsymbol{k}, t), \hat{u})$ but integrate an amplitude scaler (Section 3.1) and an embedding module (Section 3.2). These two components explicitly encode prior knowledge from harmonic analysis to improve stability, accuracy, and generalization.

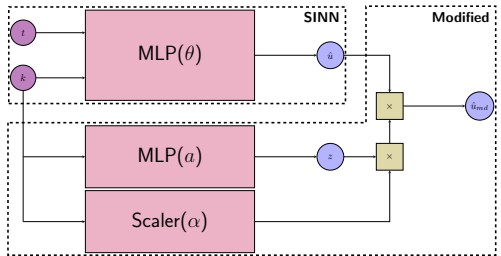

*Figure 1.* This diagram illustrates Modified SINNs. Purple circles denote input variables: $t$ and $\boldsymbol{k}$, blue circles represent output values: $\hat{u}$, $z$, and $\hat{u}_{md}$, yellow squares indicate operators ($\times$ means pointwise multiplication), and red rectangles signify neural network components where $\theta$, $a$ and $\alpha$ are the corresponding learnable parameters. The top dashed section shows the original SINNs and the bottom dashed section illustrates the modified part.

### 3.1. Coefficient Decay

The decay of expansion coefficients plays a fundamental role in spectral methods, as it directly determines the accuracy and efficiency of numerical schemes. In harmonic analysis, it is known that the rate of coefficient decay is bounded by smoothness (see Appendix E for a discussion of the decay rate for spectral bases). Since the smoothness and analyticity of PDE solutions can often be determined from the structure of the given equations, the coefficients, and the source term (Canuto et al., 2006), the decay rate of corresponding spectral coefficients can be regarded as prior knowledge when learning their distribution. Thus, in Modified SINNs, a scaling factor is applied to the output. For example, when considering the analytic functions in Fourier basis, Theorem E.1 establishes that the magnitude of the corresponding coefficients satisfies $|\hat{u}_{\boldsymbol{k}}| \leq C\, e^{-\|\boldsymbol{\alpha} \cdot \boldsymbol{k}\|_1}$ for some $\boldsymbol{\alpha}, C > 0$, thus the output of SINNs is multiplied by $e^{-\|\boldsymbol{\alpha} \cdot \boldsymbol{k}\|_1}$, where $\boldsymbol{\alpha}$ is a learnable parameter. By reweighting different frequencies according to this prior, the scaler redistributes approximation capacity toward strongly dominant low-frequency components and suppresses weakly dominant high-frequency components. When the assumed decay rate is consistent with the true regularity of the solution, this scaler improves generalization efficiently.

### 3.2. Basis Embedding

In spectral methods, the coefficient $\hat{u}_{\boldsymbol{k}}$ is calculated by integration:

$$\hat{u}_{\boldsymbol{k}} = \int_\Omega u(\boldsymbol{x})\phi_{\boldsymbol{k}}(\boldsymbol{x})\mathrm{d}\boldsymbol{x}. \qquad (12)$$

The discrete forward transform is computed via a quadrature rule on the collocation point set $\mathcal{X} \subset \Omega$:

$$\hat{u}_{\boldsymbol{k}}^{\mathcal{X}} = \sum_{\boldsymbol{x} \in \mathcal{X}} \omega(\boldsymbol{x})u(\boldsymbol{x})\phi_{\boldsymbol{k}}(\boldsymbol{x}), \qquad (13)$$

where $\omega(\boldsymbol{x})$ is the weight of the chosen quadrature. Since the basis expression is known and the function $u(\boldsymbol{x})$ can be approximated by neural networks, Modified SINNs introduce a much coarser collocation point set $\mathcal{X}_c$, where $|\mathcal{X}_c| \ll |\mathcal{X}|$, and the embedding:

$$z^a(\boldsymbol{k}) = \mathrm{MLP}\left(\boldsymbol{\phi}_{\boldsymbol{k}}\right), \qquad (14)$$

where $\boldsymbol{\phi}_{\boldsymbol{k}} = (\phi_{k_1}(x_1), \phi_{k_2}(x_2), \ldots, \phi_{k_d}(x_d))_{\boldsymbol{x} \in \mathcal{X}_c} \in \mathbb{R}^{d \times |\mathcal{X}_c|}$ is the matrix that the $i$-th row of $\boldsymbol{\phi}_{\boldsymbol{k}}$ consists of the evaluations of the one-dimensional basis function $\phi_{k_i}$ along the $i$-th coordinate at the collocation points $\mathcal{X}_c$, and $a$ represents the learnable parameters of the embedding module. The MLP is designed to approximate the underlying physical function $u(\boldsymbol{x})$; thus, the basis embedding can provide the neural network with a more powerful capability to represent oscillatory coefficients. If the parameter $a$ is large enough (which can naturally satisfy the condition $|z^a(\boldsymbol{k}) - \hat{u}_{\boldsymbol{k}}^{\mathcal{X}_c}| \ll \varepsilon(|\mathcal{X}_c|)$, for all $\boldsymbol{k}$ in the valid index set by the universal approximation theorem (Hornik et al., 1989).), Theorem 3.1 reveals that $z^a(\boldsymbol{k})$ converges to $\hat{u}_{\boldsymbol{k}}^{\mathcal{X}}$ with the same convergence rate as that of the quadrature[4].

**Theorem 3.1.** *Suppose that, for a given quadrature and a given integrand function $f(\boldsymbol{x}) = u(\boldsymbol{x})\phi_{\boldsymbol{k}}(\boldsymbol{x})$, the error of the quadrature depends on the number of collocation points:*

$$\left| \int_\Omega f(\boldsymbol{x})\mathrm{d}\boldsymbol{x} - \sum_{\boldsymbol{x} \in \mathcal{X}} \omega(\boldsymbol{x})f(\boldsymbol{x}) \right| \leq \varepsilon(|\mathcal{X}|). \qquad (15)$$

*Let $\mathcal{K}_w$, $\mathcal{K}_t$ denote the total dataset and the training dataset respectively, and $\Phi = [\boldsymbol{\phi}_{\boldsymbol{k}}]_{\boldsymbol{k} \in \mathcal{K}_t} \in \mathbb{R}^{|\mathcal{X}_c| \times |\mathcal{K}_t|}$ is the matrix stacked by vectors $\boldsymbol{\phi}_{\boldsymbol{k}}$, $\boldsymbol{k} \in \mathcal{K}_t$. If $\mathrm{Rank}(\Phi) \geq |\mathcal{X}_c|$ and $|z^a(\boldsymbol{k}) - \hat{u}_{\boldsymbol{k}}^{\mathcal{X}_c}| \ll \varepsilon(|\mathcal{X}_c|)$, $\forall \boldsymbol{k} \in \mathcal{K}_t$, then*

$$|z^a(\boldsymbol{k}) - \hat{u}_{\boldsymbol{k}}^{\mathcal{X}}| = \mathcal{O}\left(\varepsilon(|\mathcal{X}_c|)\right), \quad \forall \boldsymbol{k} \in \mathcal{K}_w \setminus \mathcal{K}_t. \qquad (16)$$

### 3.3. Integrated Architecture

Combined with the learnable scaler in Section 3.1 and embedding module in Section 3.2, for the smooth solution $u(\boldsymbol{x}, t)$, the output $\hat{u}_{md}^{\tilde{\theta}}$ of modified SINNs based on Fourier basis can be represented as:

$$\hat{u}_{md}^{\tilde{\theta}}(\boldsymbol{k}, t) = z^a(\boldsymbol{k})\hat{u}^\theta(\boldsymbol{k}, t)e^{-\|\boldsymbol{\alpha} \cdot \boldsymbol{k}\|_1}, \qquad (17)$$

where $a, \theta, \boldsymbol{\alpha}$ are learnable parameters and $\tilde{\theta} = \{a, \theta, \boldsymbol{\alpha}\}$ is the collection. Figure 1 concludes the Modified SINNs

---

[4]The proof of Theorem 3.1 and further discussions, including practical implementation, are provided in Appendix F.

compared with SINNs. In Figure 1, MLP($a$) denotes the implementation of $z^a(\boldsymbol{k})$, realized by the MLP architecture, and Scaler($\alpha$)[5] represents the implementation function of $e^{-\|\boldsymbol{\alpha} \cdot \boldsymbol{k}\|_1}$. We also provide a convergence analysis in Appendix G.

*Remark* 3.1. (i) If the used basis is not Fourier basis, then one can use the coefficient decay theorem of other spectral bases as discussed in Appendix E and change the function $\phi$ in $z^a(\boldsymbol{k})$. (ii) If the solution is not smooth, then the output is

$$\hat{u}_{md}^{\bar{\theta}}(\boldsymbol{k}, t) = z^a(\boldsymbol{k})\hat{u}^\theta(\boldsymbol{k}, t) / \prod_{j=1}^d |k_j|^{n_j}, \qquad (18)$$

where $\boldsymbol{n}$ uses the same notation as in Theorem E.1 and is learnable. Notably, the decay scaling is applied only to nonzero frequency components; for $k_j = 0$, the corresponding factor is set to one to avoid singularity.

The coefficient decay scaler explicitly encodes the analytical prior that spectral coefficients diminish with increasing frequency. By enforcing a learnable decay profile, the network produces coefficients with physically plausible magnitudes across the entire spectral domain, preventing overestimation in high-frequency regions and stabilizing training, especially when the sampled frequencies are sparse. The basis embedding module incorporates the low-dimensional structural information carried by the spectral basis itself, providing the network with an explicit representation of this structure. As a result, Modified SINNs are able to generalize to missing frequencies outside the training set by leveraging the essential prior knowledge in amplitude and structure. We provide further discussion and experiments in Section 4.1 and Appendix I.

In high-dimensional problems, the principal challenge lies in identifying valid coefficients[6]. Existing research typically addresses this difficulty by either restricting the index set to a predetermined structure, such as a hyperbolic cross, or by adaptively identifying coefficients based on properties of the underlying equation (see, e.g., (Gross & Iwen, 2025)). However, because identification of all valid coefficients becomes computationally prohibitive as the dimension increases, accurately approximating these missing valid coefficients can substantially reconstruct the complete set of valid coefficients and further improve the accuracy of SGSM.

### 3.4. Genuinely Dense Coefficients in High Dimensions

Before presenting the experiments, this section discusses genuinely dense coefficients in high dimensions, since all experiments in this work are conducted under sparse coeffi-

cient settings. Here, "genuinely dense coefficients" means that the target function admits no sparse representation under any basis. To the best of our knowledge, PDEs with genuinely dense coefficients remain fundamentally challenging for existing high-dimensional methods, including PINNs and DRMs.

Take PINNs as an example. Let $f(\boldsymbol{x}) : \mathbb{R}^d \to \mathbb{R}$ denote the target function and let $u^\theta(\boldsymbol{x})$ be the neural network approximation. Since the universal approximation theorem shows that MLPs can approximate arbitrary continuous functions, including genuinely dense ones, we consider an MLP here. For a standard MLP, if the outputs of the penultimate layer are denoted by $\{\phi_j(\boldsymbol{x})\}_{j=1}^w$, then the final network output can be written as

$$u^\theta(\boldsymbol{x}) = \sum_{j=1}^w c_j \phi_j(\boldsymbol{x}), \qquad (19)$$

where $c_j$ are the coefficients in the final linear layer (the bias term can be absorbed by including $\phi \equiv 1$). Therefore, even a neural network can be interpreted as representing the target function by the MLP-constructed basis $\{\phi_j(\boldsymbol{x})\}_{j=1}^w$. Then, if the target function is genuinely dense, accurate approximation in high dimensions requires either an extremely large width $w$ or otherwise incurs a large approximation error. This suggests that PDEs with genuinely dense coefficients remain highly challenging for PINNs.

## 4. Experiments

In this section, experiments are conducted to show the performance of SINNs (here and in the following, "SINNs" refers to our Modified SINNs) and compare them with SGSM and PINNs. Since (Yu et al., 2025) has already compared SINNs with other SOTA methods on low-dimensional PDEs ($d \leq 3$), the experiments in this section mainly focus on $d > 3$. First, we conduct experiments on approximating missing coefficients in Section 4.1 to demonstrate this capability. Second, we use some middle-dimensional problems to compare with SGSM in Section 4.2. Third, we conduct experiments on high-dimensional problems to compare with PINNs and DRMs in Section 4.3. Fourth, we explore the strategy that combines spectral methods and SINNs in Section 4.4. Finally, we conduct an ablation study to demonstrate the improved stability and accuracy of the proposed two modules in Section 4.5. Notably, since PINNs and DRMs are unable to recover missing coefficients and generally perform worse than spectral methods in middle-dimensional problems, they are not considered baselines in Section 4.2, however, we provide their performance in Table 10 as valuable empirical results. Also, as we demonstrate that SGSM is not scalable to high-dimensional problems, it is not included in the comparisons presented in Section 4.3.

We mainly focus on the Fourier basis (representative of

---

[5]Here we use $\alpha$ rather than $\boldsymbol{\alpha}$ to denote the learnable parameters in networks

[6]A valid coefficient means its magnitude is greater than a given threshold (e.g., $10^{-6}$).

diagonal differential operators) and the Chebyshev basis (representative of non-diagonal differential operators). As typical representatives of two distinct categories of spectral bases, our methodologies and results are applicable to analogous spectral bases. The details of all experiments are provided in Appendix J. Since both the forward and inverse transforms of spectral methods suffer from CoD for high-dimensional functions, we discuss the algorithms and costs of both transforms in Appendix J.4. Furthermore, for non-linear equations, since generating high-accuracy numerical targets for high-dimensional nonlinear PDEs is difficult, we conduct an experiment on the 2D Navier–Stokes equations to demonstrate the feasibility of approximating missing coefficients for nonlinear equations in Appendix K.

The metric used in this paper is the relative $L^2$ error (see Appendix M). These experiments are conducted on an A40-40G with CUDA version 12.4. Other hyperparameters refer to the GitHub repository https://github.com/DUCH714/SINN_high/tree/master.

### 4.1. Approximate Missing Coefficients

First, we evaluate the network's ability to approximate randomly masked coefficients sampled from a uniform distribution. Although such masking is unrealistic since significant coefficients are usually concentrated at low frequencies, it provides an intuitive benchmark for evaluating the performance of Modified SINNs. A more practical experiment is conducted in Section 4.4.

We set $1 - s$ to be the proportion of missing coefficients. Let $N_{\text{valid}}$ represent the total number of valid coefficients. We randomly sample $sN_{\text{valid}}$ coefficients to form the training set. We demonstrate the performance in Table 1 with Fourier basis and $s = 0.9$. In Table 1, "Error" denotes the relative $L^2$ error over all valid coefficients, "Error (Missing)" denotes the relative $L^2$ error over the missing coefficients, and "Rate" denotes the training rate of SINNs. The details of this experiment, and other additional experiments (e.g., Chebyshev basis) are provided in Appendix I.

The results in Table 1 show that SINNs can accurately approximate masked Fourier coefficients when only a partial spectral set ($s = 0.9$) is available. Across all dimensions, the error on the missing coefficients ("Error (Missing)") is comparable to the overall error ("Error"), indicating that coefficient masking is not the dominant source of approximation error. Instead, the errors are mainly driven by the intrinsic oscillatory complexity of the solution. These results confirm that SINN effectively exploits spectral correlations to recover missing coefficients. Furthermore, to explore the limitations of approximating missing coefficients, we conduct experiments with different $s$ and $|\mathcal{X}_c|$ on the 2D Chebyshev basis. The results are shown in Figure 2 and indicate that an approximation accuracy of

$\mathcal{O}(10^{-3})$ can still be achieved when $s \geq 0.6$. Additionally, Figure 3 demonstrates the approximation performance for the 2D Chebyshev basis with the function $u(x,y) = e^{-5(x^2+y^2)} \sin(5x)\sin(3y) + 0.2\cos(10xy)$, $x,y \in [0,1]$ where 10% of the coefficients are randomly sampled to be masked. The results show that the predicted coefficients are close to the target values, and the errors are mainly concentrated in large magnitude coefficients.

*Table 1.* Predicting missing coefficients for Fourier basis with $s = 0.9$.

| DIM | ERROR | ERROR (MISSING) | RATE (ITE/S) |
|---|---|---|---|
| 2 | 4.21E-4 ± 8.00E-5 | 1.15E-3 ± 1.20E-3 | 1.16E+3 |
| 3 | 5.60E-5 ± 1.40E-5 | 7.14E-5 ± 4.27E-6 | 8.01E+2 |
| 6 | 2.40E-2 ± 3.23E-2 | 3.08E-2 ± 2.76E-2 | 6.43E+1 |
| 8 | 3.92E-3 ± 2.42E-4 | 4.98E-3 ± 3.54E-4 | 5.34E+1 |

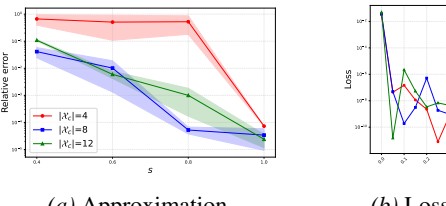
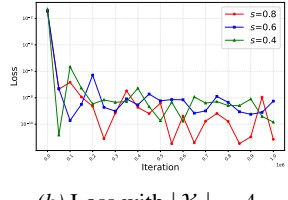

*(a)* Approximation      *(b)* Loss with $|\mathcal{X}_c| = 4$

*Figure 2.* Approximation performance for missing coefficients on the Chebyshev basis ($d = 2$) with different $s$ and $|\mathcal{X}_c|$. Figure a shows the approximation relative error. Figure b shows the evolution of the training loss for $s = 0.8, 0.6, 0.4$ with $|\mathcal{X}_c| = 4$.

### 4.2. Compare with Spectral Methods

Since the SGSM can only handle PDEs with $d \lesssim 10$, we conducted experiments on several types of PDEs[7]: the Poisson's equation and heat equation with the Fourier basis for $d = 2, 3, 5, 8$, as well as the steady- and time-dependent convection equations with the Chebyshev basis for $d = 2, 3, 5, 6$. To fairly compare the performance of SINNs and SGSM ($q = 10$), we use the same masked matrix for both methods.

Figure 4 shows that when $s = 1$, SGSM achieves higher accuracy due to its high-accuracy computation of spectral coefficients. However, when coefficients become partially unavailable ($s < 1$), the performance of SGSM deteriorates rapidly, while SINNs deteriorate more slowly. These results indicate the advantage of SINNs over SGSM when some coefficients are missing, demonstrating their ability to effectively approximate unknown spectral coefficients and highlighting the robustness of SINNs in incomplete spectral cases.

---

[7]See Appendix J for details of these PDEs.

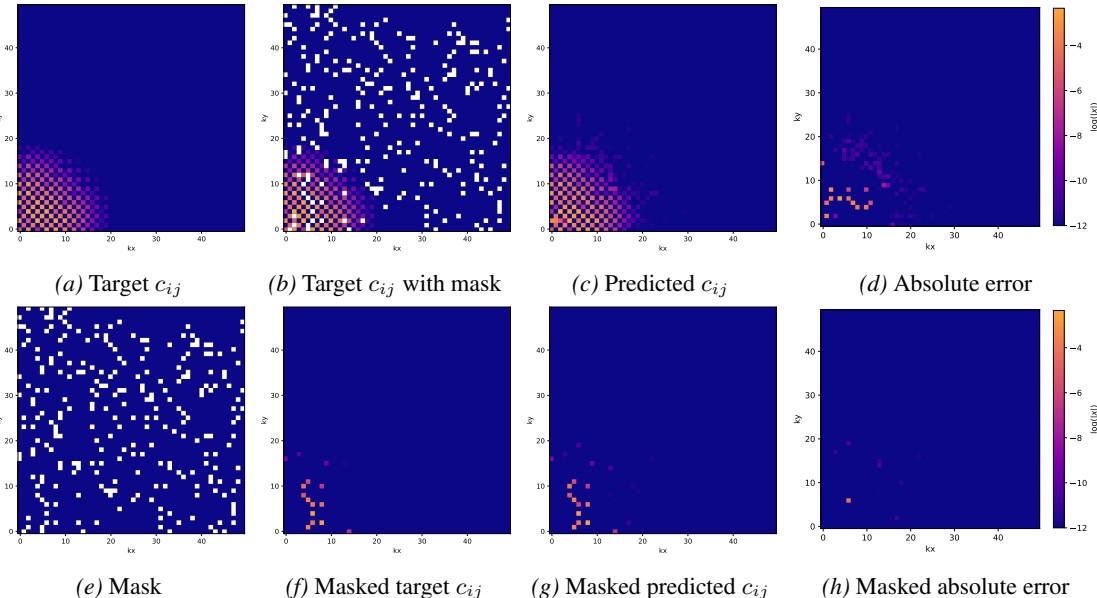

*(a)* Target $c_{ij}$     *(b)* Target $c_{ij}$ with mask     *(c)* Predicted $c_{ij}$     *(d)* Absolute error

*(e)* Mask     *(f)* Masked target $c_{ij}$     *(g)* Masked predicted $c_{ij}$     *(h)* Masked absolute error

*Figure 3.* The approximation with 2D Chebyshev polynomials. a, c, d show the distribution of target values, predicted values and the absolute errors respectively; f, g and h highlight the masked elements respectively. The axes $k_x, k_y$ denote the corresponding frequency of 2D Chebyshev polynomials. The masked matrix masks 10% of the original matrix and is generated by uniformly sampling. b and e show the distribution of masked elements (the white pixels). All figures are filled by logarithm of the absolute values.

## 4.3. High-dimensional PDEs

For high-dimensional PDEs, we compared our SINNs with representative machine learning models: physics-informed neural networks (PINNs) (Raissi et al., 2019) and the deep Ritz methods (DRMs) (Weinan & Yu, 2018) on high-dimensional Poisson's equations.

For PINNs, we use the network physics-informed residual adaptive networks (PirateNets) (Wang et al., 2024) which can mitigate the spectral bias for PINNs. Notably, for the standard deviation $s$ of Fourier feature embeddings in PirateNets, we set $s = 1/d$ to avoid the accumulation problem when $d$ is large. For DRMs, in addition to periodic embedding (Wang et al., 2024), we use natural deep Ritz methods (NDRMs) (Yu & Zhang, 2025), which reveal the sensitivity to the weight of the penalty term ($\mathcal{L}_b$) and propose integrating the penalty term into the energy functional loss. However, since curl $\boldsymbol{u}$ is computationally expensive for high-dimensional functions, we omit the proposed correction term of NDRMs. The formulations of the models used are introduced in Appendix L.

Tables 2 and 3 demonstrates the results for varying dimensionalities. While the errors of PINNs and DRMs increase rapidly with dimensionality, SINNs maintain significantly lower errors, especially in high dimensions. These results highlight the advantage of SINNs in high-dimensional problems. Furthermore, we conducted additional experiments on challenging problems to demonstrate the advantage of SINNs in multiscale problems, the results are shown in

Table 11.

*Table 2.* High-dimensional Poisson's equations.

| DIM | PINN | DRM | SINN |
|-----|------|-----|------|
| 2 | 3.20E-4 ± 6.77E-5 | 2.48E-3 ± 4.36E-5 | 1.61E-4 ± 1.64E-5 |
| 5 | 5.54E-3 ± 1.13E-3 | 7.95E-3 ± 1.87E-4 | 2.24E-4 ± 6.80E-5 |
| 10 | 9.68E-3 ± 1.34E-3 | 1.03E-2 ± 4.82E-4 | 1.99E-4 ± 3.97E-5 |
| 30 | 1.70E-2 ± 2.20E-3 | 2.23E-2 ± 1.45E-3 | 2.76E-4 ± 8.51E-5 |
| 50 | 3.17E-2 ± 5.13E-3 | 3.20E-2 ± 5.19E-3 | 1.20E-3 ± 6.20E-4 |
| 100 | 5.50E-1 ± 7.46E-1 | 5.94E-2 ± 1.51E-2 | 7.75E-3 ± 1.67E-3 |

*Table 3.* High-dimensional Heat equations.

| DIM | PINN | DRM | SINN |
|-----|------|-----|------|
| 2 | 8.61E-4 ± 3.76E-4 | 1.12E-1 ± 4.10E-2 | 2.31E-4 ± 2.00E-5 |
| 5 | 1.41E-2 ± 2.00E-3 | 5.33E-2 ± 2.73E-2 | 2.20E-4 ± 4.53E-5 |
| 10 | 2.09E-2 ± 1.63E-3 | 3.40E-2 ± 1.27E-2 | 3.66E-4 ± 3.80E-5 |
| 30 | 3.01E-2 ± 5.57E-3 | 3.10E-2 ± 1.36E-2 | 1.46E-3 ± 4.30E-4 |
| 50 | 5.86E-2 ± 6.25E-3 | 2.02E-2 ± 8.00E-3 | 8.57E-3 ± 3.17E-3 |
| 100 | 7.39E-1 ± 1.99E-2 | 4.18E-1 ± 5.60E-1 | 3.69E-1 ± 4.47E-1 |

## 4.4. Combine SGSM and SINNs

Figure 4 also shows that, when $s = 1$, SGSM achieves higher accuracy than SINNs due to its highly accurate computation of frequency coefficients. This section proposes a hybrid strategy in which SGSM is used to compute the

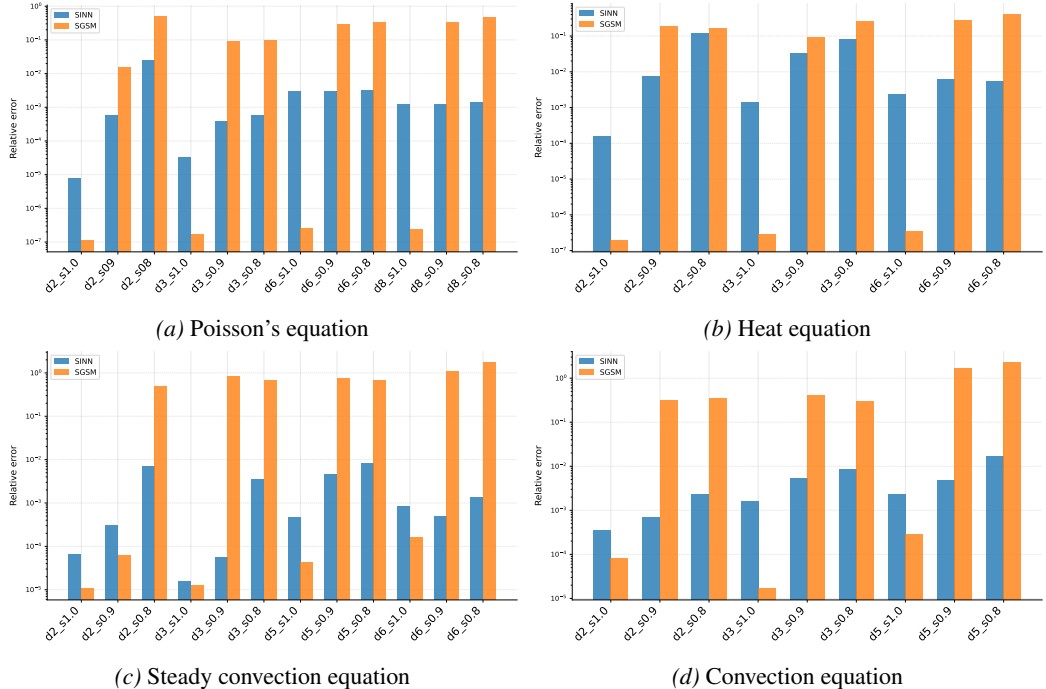

*(a) Poisson's equation*  *(b) Heat equation*

*(c) Steady convection equation*  *(d) Convection equation*

*Figure 4.* Compare SGSM and SINN in different problems on varying dimensions $d$ and the proportion of available coefficients $s$. In figures, $dX_sY$ denotes $d = X$ and $s = Y$.

coefficients corresponding to unmasked frequencies, while SINNs are trained on these SGSM-computed coefficients to predict the coefficients corresponding to masked frequencies. The experiments in this section are conducted on the time-dependent harmonic oscillator Schrödinger equation benchmark taken from (De la Fuente & Lamorena, 2024). Notably, to simulate a practical experimental setting, the missing frequencies in this study are sampled from a distribution with higher density over high-frequency components.

The results in Table 4 show that the proposed SGSM+SINN strategy significantly improves the accuracy of SGSM when $s < 1$. While SGSM exhibits a degradation in accuracy as $s$ decreases, incorporating SINNs recovers the masked spectral coefficients and reduces the error by up to several orders of magnitude across all tested dimensions.

### 4.5. Ablation Study

Here we present an ablation study of our proposed Modified SINNs on the 2D Fourier basis. We provide the results in Table 5, where 'w/o' means 'without' and the label 'w/o Decay & Embedding' exactly represents the original SINN architecture.

The results demonstrate that incorporating either the coefficient decay module or the embedding module yields consistent performance improvements; integrating both (i.e., using the full Modified SINN) delivers the best performance while retaining a comparable iteration speed. In particular,

*Table 4.* High-dimensional Schrödinger equations solved by SGSM+SINN.

| METRIC | DIMENSION | | |
|---|---|---|---|
| | 2 | 5 | 8 |
| SGSM ($s = 1.0$) | 2.62E-7 | 4.42E-6 | 3.89E-4 |
| $N_{\text{VALID}}$ | 1.47E+3 | 1.91E+5 | 8.46E+5 |
| SGSM ($s = 0.9$) | 2.28E-3 | 9.37E-2 | 3.05E-1 |
| +SINN | 8.83E-5 | 1.60E-2 | 1.23E-3 |
| PROMOTION | 96.13% | 82.92% | 99.60% |
| $N_{\text{VALID}}$ | 1.32E+3 | 1.72E+5 | 7.62E+5 |
| SGSM ($s = 0.8$) | 2.07E-1 | 2.43E-1 | 4.41E-1 |
| +SINN | 1.10E-3 | 3.74E-2 | 1.67E-3 |
| PROMOTION | 99.47% | 84.61% | 99.62% |
| $N_{\text{VALID}}$ | 1.18E+3 | 1.53E+5 | 6.77E+5 |

for $s = 0.6$, the coefficient decay module slightly improves the prediction of missing coefficients, while the embedding module significantly enhances the accuracy. The results reveal that the coefficient decay module primarily reduces numerical error, whereas the embedding module, which implicitly reconstructs the physical function, is more effective in approximating the missing-coefficient error.

*Table 5.* Ablation Study

| ARCHITECTURE | $s = 1.0$ ERROR | $s = 0.6$ ERROR | RATE (ITE/S) |
|---|---|---|---|
| W/O DECAY & EMBEDDING | 6.54E-3 ± 1.17E-3 | 2.11E-1 ± 5.37E-2 | 1.60E+3 |
| W/O EMBEDDING | 1.09E-3 ± 1.73E-4 | 2.06E-1 ± 1.22E-2 | 1.59E+3 |
| W/O DECAY | 1.25E-3 ± 3.44E-4 | 3.27E-3 ± 1.61E-3 | 1.56E+3 |
| MODIFIED SINN | 5.55E-4 ± 7.05E-5 | 3.08E-3 ± 1.53E-3 | 1.56E+3 |

## 5. Conclusion

We propose Modified Spectral-Informed Neural Networks (Modified SINNs) for solving high-dimensional PDEs by integrating prior knowledge into SINNs. Modified SINNs achieve improved stability and accuracy and enable effective approximation of unknown spectral coefficients. Experiments demonstrate that Modified SINNs achieve superior accuracy compared to machine learning methods on high-dimensional PDEs and outperform spectral methods on middle-dimensional PDEs with incomplete spectral coefficients.

**Limitations and future research.** All experiments above are conducted under the assumption of sparse coefficients. However, existing approaches for high-dimensional problems, including PINNs, fundamentally rely on the presence of special low-dimensional structures in high-dimensional functions (Trefethen, 2017), and these structures can induce sparsity in the spectral coefficients under suitable bases. As discussed in Section 3.4, high-dimensional problems with genuinely dense coefficients and without any underlying low-dimensional structure are intractable for all current methods.

Although Modified SINNs can approximate unknown coefficients, the highly accurate results are achieved when the solution has specific structures, e.g., symmetry and separability. In our experiments, except Appendix I.1, Appendix K, and Appendix J.2, the remaining cases use solutions with separable or symmetric structures. Experiments with specific structures generally achieve higher accuracy than those without, suggesting that the approximation of unknown coefficients benefits significantly from structural information in the solution.

Furthermore, the approximation may fail in some extreme situations, for example, when the missing coefficients concentrate in a specific range or when too many low frequencies are masked. However, in practical applications, low-frequency coefficients are typically unmasked, as these components are often regarded as contributing most significantly to the function. Consequently, an important direction for future research is the identification of unmasked frequencies, for which some algorithms have been proposed for specific problems (Gross & Iwen, 2025; Kämmerer et al., 2022). In this direction, machine learning has not yet been widely applied and may be leveraged to infer potential structural

regularities of given PDEs and to more accurately identify the most informative spectral coefficients.

## Impact Statement

This paper presents work whose goal is to advance the field of Machine Learning. There are many potential societal consequences of our work, none of which we feel must be specifically highlighted here.

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

## A. Spectral Bases

Here we briefly present some representative 1D spectral bases, from which the multidimensional basis can be constructed via tensor products.

1. Fourier basis: $\phi_n(x) = e^{inx}$ for $n \in \mathbb{Z}$, defined on $x \in [0, 2\pi]$. i denotes the imaginary unit.

2. Sine basis: $\phi_n(x) = \sin(nx)$ for $n \geq 1$, defined on $x \in [0, \pi]$.

3. Cosine basis: $\phi_n(x) = \cos(nx)$ for $n \geq 0$, defined on $x \in [0, \pi]$.

4. Chebyshev polynomials (first kind): $T_n(x) = \cos(n \arccos x)$ for $n \geq 0$, defined on $x \in [-1, 1]$.

5. Legendre polynomials: $P_n(x) = \frac{1}{2^n n!} \frac{d^n}{dx^n} \left[ (x^2 - 1)^n \right]$ for $n \geq 0$, defined on $x \in [-1, 1]$.

6. Hermite polynomials: $H_n(x) = (-1)^n e^{x^2} \frac{d^n}{dx^n} \left[ e^{-x^2} \right]$ for $n \geq 0$, defined on $x \in \mathbb{R}$.

7. Laguerre polynomials: $L_n(x) = \frac{1}{n!} e^x \frac{d^n}{dx^n} \left[ x^n e^{-x} \right]$ for $n \geq 0$, defined on $x \in [0, \infty)$.

Among these spectral bases, the Fourier basis yields a diagonal representation of differential operators, and the sine and cosine bases yield diagonal representations for second-order operators. In contrast, the remaining bases have non-diagonal differential operators, as differentiation induces coupling among spectral modes.

## B. Formulations of Spectral Methods

Different formulations of spectral methods arise depending on how the residual, i.e.,

$$R_N(\boldsymbol{x}) = \mathcal{N}[u_N](\boldsymbol{x}) - f(\boldsymbol{x}), \tag{20}$$

is enforced. Among the most widely used are the Galerkin and collocation approaches:

- **Galerkin method:** In the Galerkin formulation, one requires the residual to be orthogonal to the approximation space spanned by the chosen basis functions. More precisely, for each basis function $\phi_{\boldsymbol{k}}$ in the finite-dimensional space $\mathcal{K}_N$, one enforces

$$\int_\Omega R_N(\boldsymbol{x}) \, \phi_{\boldsymbol{k}}(\boldsymbol{x}) \, d\boldsymbol{x} = 0, \quad \forall \boldsymbol{k} \in \mathcal{K}_N. \tag{21}$$

  The Galerkin method arises naturally from variational principles and functional analysis, making it closely related to weak formulations of PDEs. It is widely used in problems where conservation laws, stability, or energy estimates are critical, such as fluid dynamics and structural mechanics. By construction, the Galerkin formulation often inherits stability properties from the underlying PDE, particularly when symmetric or self-adjoint operators are involved (Orszag, 1971a; Canuto et al., 2007).

- **Collocation method:** In contrast, the collocation (or pseudo-spectral) method enforces the residual to vanish exactly at the collocation points $\{\boldsymbol{x}_i\}_{i=1}^N$:

$$\mathcal{N}[u_N](\boldsymbol{x}_i) = f(\boldsymbol{x}_i), \quad i = 1, \dots, N. \tag{22}$$

  The collocation method is grounded in interpolation theory, particularly polynomial interpolation at special nodes such as Chebyshev or Legendre points. It is especially popular in practice due to its straightforward implementation and efficiency, making it suitable for time-dependent problems, nonlinear PDEs, and high-dimensional approximations. The stability of collocation methods strongly depends on the choice of collocation nodes; Chebyshev or Legendre nodes yield well-conditioned systems and spectral accuracy for smooth problems, whereas equispaced nodes may lead to instability due to Runge's phenomenon (Trefethen, 2000).

Both formulations share the hallmark feature of spectral methods, namely their ability to achieve extremely fast convergence rates for smooth or analytic solutions. However, they differ in implementation, theoretical properties, and numerical stability, which motivates choosing one formulation over the other depending on the application. The SINNs are inspired by the collocation methods.

Additionally, for time-dependent PDEs, the temporal variable is generally discretized using high-order finite difference methods, such as Runge–Kutta methods, to ensure that the temporal accuracy matches the spatial accuracy of spectral methods (Tang, 2006).

In this paper, the numerical experiments use collocation-based spectral methods for spatial variables and the third-order total variation diminishing Runge–Kutta method (third-order TVD RK) for temporal variables in Section 4.2 for SGSM.

## C. Hyperbolic Cross

The hyperbolic cross approximation is a fundamental tool in multivariate numerical analysis. The main idea is to reduce the set of multi-indices $\mathcal{K}_N$ to the so-called hyperbolic cross index set. For classes of multivariate functions ($d \geq 3$) with mixed smoothness, hyperbolic cross approximation achieves superior error convergence, whereas full-grid approximation exhibits significant degradation in convergence due to the curse of dimensionality (Dũng et al., 2018). To further explain how hyperbolic cross helps SGSM solve high-dimensional problems, we define the hyperbolic cross index set:

$$\Upsilon_N^H = \left\{ \boldsymbol{k} \in \mathbb{N}_0^d : 1 \leq |\boldsymbol{k}|_{\mathrm{mix}} \leq N \right\}, \tag{23}$$

where $|\boldsymbol{k}|_{\mathrm{mix}} = \prod_{j=1}^d \max\{1, k_j\}$. This construction effectively favors basis functions with balanced contributions across dimensions, while discarding those dominated by high frequencies in multiple coordinates simultaneously. As a result, $\Upsilon_N^H$ achieves near-optimal approximation rates for functions with mixed regularity, while the number of nodes grows only polynomially with $N$ and only mildly with the dimension $d$. In this way, the hyperbolic cross approximation provides a principled reduction of degrees of freedom that preserves accuracy in high-dimensional spectral methods. However, as noted in (Shen & Yu, 2010), the classical hyperbolic cross approximation is of limited practicality because it lacks a direct representation in the physical space. The authors further demonstrated that, in the frequency domain, the hyperbolic cross approximation can be closely linked to Smolyak's sparse grid through an interpolation operator constructed from hierarchical basis functions. First, we introduce the notation used in the following:

$$\begin{aligned}
\mathcal{X}_d^q &:= \bigcup_{d \leq |\boldsymbol{i}|_1 \leq q} \mathcal{X}^{i_1} \times \mathcal{X}^{i_2} \times \cdots \times \mathcal{X}^{i_d}, \\
\mathcal{I}_d^q &:= \bigcup_{d \leq |\boldsymbol{i}|_1 \leq q} \tilde{\mathcal{I}}^{i_1} \times \tilde{\mathcal{I}}^{i_2} \times \cdots \times \tilde{\mathcal{I}}^{i_d},
\end{aligned} \tag{24}$$

where $\mathcal{X}^i = \{x_j^i\}_{j=0}^{N_i}$, $\tilde{\mathcal{I}}^i = \mathcal{I}^i \backslash \mathcal{I}^{i-1}$, and $\mathcal{I}^i = \{0, 1, \cdots, N_i - 1\}$. The index set $\mathcal{I}_d^q$ has a close relationship with $\Upsilon_N^H$ and can be regarded as a quasi-hyperbolic cross index set. To provide an intuitive understanding, we show the distribution of $\mathcal{I}_d^q$ in Figure 5 for $d = 2$ and Figure 6 for $d = 3$. Let $\otimes$ denote the tensor product. The Smolyak formula is defined by $\mathcal{A}(q, d) = \sum_{|\boldsymbol{i}|_1 \leq q} \left( \Delta^{i_1} \otimes \cdots \otimes \Delta^{i_d} \right)$, where $\Delta^i = \mathcal{U}^i - \mathcal{U}^{i-1}, \mathcal{U}^i(f)(x) = \sum_{k \in \mathcal{I}^i} b_k \phi_k(x)$, and $b_k$ is the coefficient. (Barthelmann et al., 2000) shows that the complexity of $\mathcal{A}(q, d)$ can be simplified to $\mathcal{O}\left( \sum_{q-d < |\boldsymbol{i}|_1 \leq q} N_{i_1} N_{i_2} \cdots N_{i_d}, \right)$.

Although the operator constructed by Smolyak's algorithm can alleviate the CoD, it still exhibits exponential growth in complexity with respect to $d$. Based on hierarchical basis, an efficient algorithm proposed in (Shen & Yu, 2010) can avoid the CoD when the cost of one-dimensional transform is $\mathcal{O}(N)$ or $\mathcal{O}(N \log N)$. Nevertheless, (Shen et al., 2011) proved that the hyperbolic cross index set satisfies $|\Upsilon_N^H| = CN(\ln N)^{d-1}$, which implies that the generation of $\Upsilon_N^H$ and hence its variant $\mathcal{I}_d^q$, still suffers from the CoD. We also confirm this problem by experiments in Appendix D. In this paper, the numerical experiments use the SGSM from (Shen & Yu, 2010) with a given $q$.

## D. Cost of generating the index set

Here we focus on generating the index set $\mathcal{I}_d^q$, which contains all valid frequency indices for the spectral basis. We argue that although the index set $\mathcal{I}_d^q$ is generated by an efficient recursive method, the cost is still significant for high-dimensional problems. Here we conduct experiments to illustrate this phenomenon in Figure 7. These experiments reveal that:

1. For fixed $d$, in low-dimensional (i.e., $d = 2, 3$), the memory usage and computation time grow sub-exponentially with $q$. However, once the dimensionality $d$ is greater than 5, the computation time grows exponentially with $q$ and the memory usage grows super-exponentially with $q$.

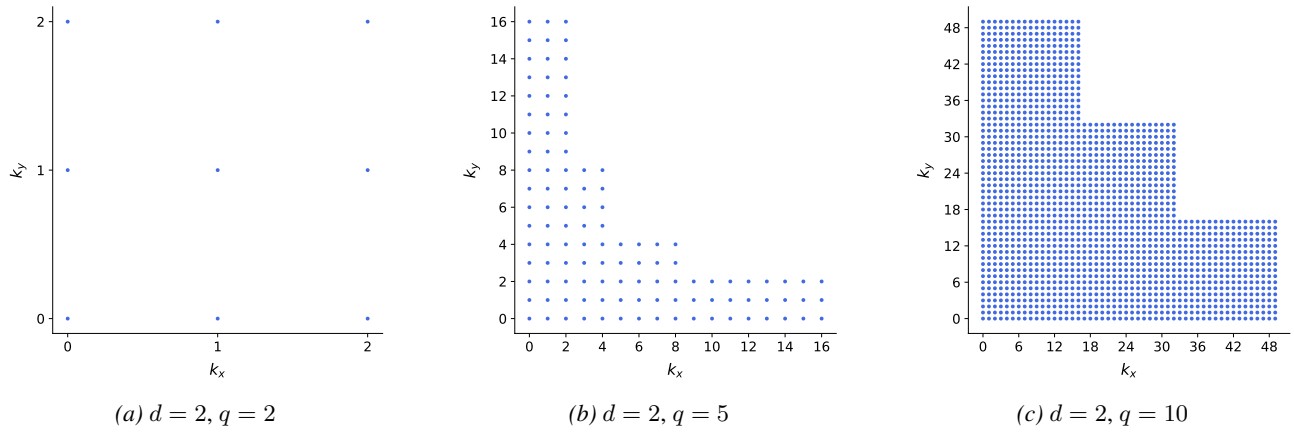

*(a) $d = 2, q = 2$*     *(b) $d = 2, q = 5$*     *(c) $d = 2, q = 10$*

*Figure 5.* Hyperbolic cross for $d = 2$ with different q values

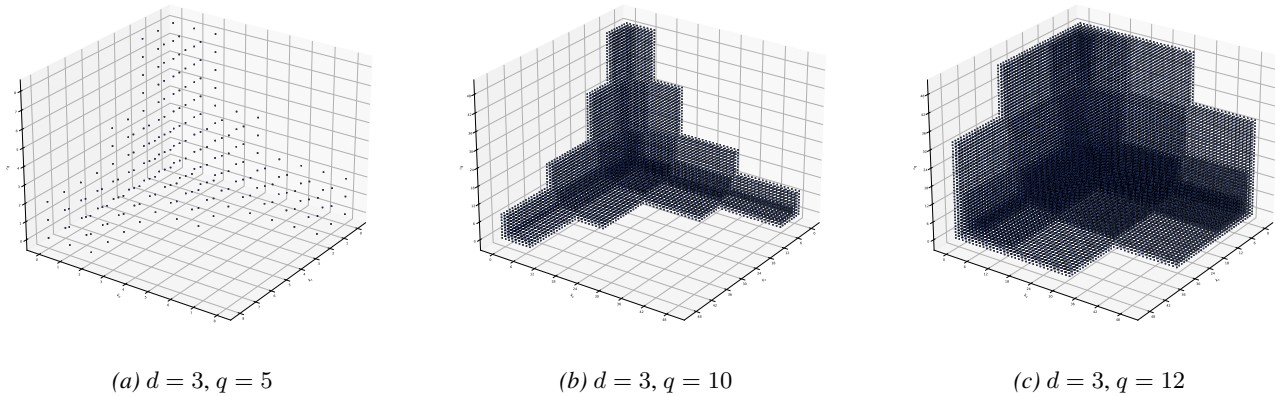

*(a) $d = 3, q = 5$*     *(b) $d = 3, q = 10$*     *(c) $d = 3, q = 12$*

*Figure 6.* Hyperbolic cross for $d = 3$ with different q values

2. For fixed $q$, the memory usage and computation time grow sub-exponentially in $d$ for small $d$, and super-exponentially for large $d$.

3. For $d > 10$, since $q > d$, the computation is infeasible due to prohibitive memory usage. Thus, the SGSM is unsuitable for high-dimensional PDEs.

Herein, while the SGSM demonstrates excellent accuracy and efficiency for low-dimensional PDEs, it becomes increasingly impractical as the problem dimensionality grows. In contrast, physics-informed neural networks (PINNs) have been shown to successfully solve PDEs with up to millions of dimensions (Shi et al., 2024), highlighting the limited competitiveness of SGSM for high-dimensional problems.

## E. Coefficient Decay for Spectral Bases

In this section, we discuss the rate of coefficient decay. First, suppose that $\mathcal{K}^* := \{ \boldsymbol{k} \in \mathcal{K} : k_j \neq 0, \ j = 1, \ldots, d \}$.

**Theorem E.1.** *Let $u : [0, 2\pi]^d \to \mathbb{R}$ be $2\pi$-periodic in each variable, $\mathrm{i}$ is the imaginary unit and let $u(\boldsymbol{x}) = \sum_{\boldsymbol{k} \in \mathcal{K}} \hat{u}_{\boldsymbol{k}} e^{\mathrm{i}\boldsymbol{k} \cdot \boldsymbol{x}}$, where $\hat{u}_{\boldsymbol{k}}$ is the Fourier coefficient. Let $\boldsymbol{k} = \{k_i\}_{i=1}^d \in \mathcal{K}$ and $\boldsymbol{n} = \{n_j\}_{j=1}^d \in \mathbb{N}^d$. Suppose that for every multi-index $\boldsymbol{\gamma} = (\gamma_1, \ldots, \gamma_d)$ satisfying $0 \leq \gamma_j \leq n_j \ \forall j = 1, \cdots, d$, the mixed derivative $\partial^{\boldsymbol{\gamma}} u$ belongs to $L^1([0, 2\pi]^d)$. Then*

$$|\hat{u}_{\boldsymbol{k}}| \leq \frac{C}{\prod_{j=1}^d |k_j|^{n_j}}, \quad \forall \boldsymbol{k} \in \mathcal{K}^* \tag{25}$$

*where $C$ is a constant that depends on $u$.*

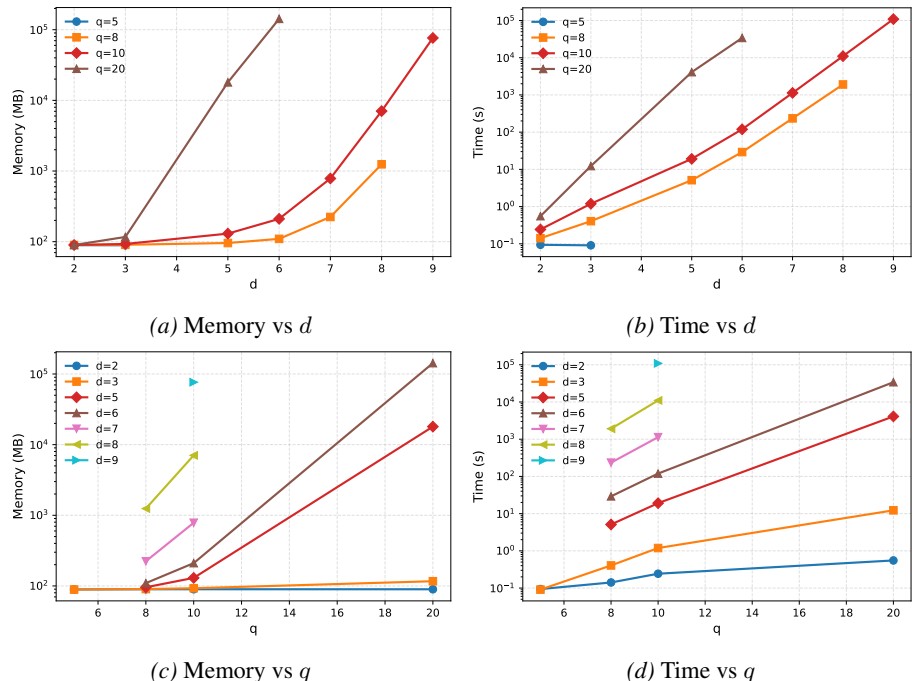

*(a)* Memory vs $d$         *(b)* Time vs $d$

*(c)* Memory vs $q$         *(d)* Time vs $q$

*Figure 7.* Performance comparison in terms of memory usage and computation time with $d$ and $q$ in $\mathcal{I}_d^q$.

*Furthermore, if $u$ is analytic,[8] then*

$$|\hat{u}_{\boldsymbol{k}}| \leq C\, e^{-\|\boldsymbol{\alpha}\cdot\boldsymbol{k}\|_1}, \qquad \forall \boldsymbol{k} \in \mathcal{K}, \tag{26}$$

*where $C, \boldsymbol{\alpha} = \{\alpha_i\}_{i=1}^d$ are constants that depend on $u$.*

For the Fourier basis, Theorem E.1 reveals the relationship between decay rate and smoothness. There is no proof of this theorem since this theorem merely extends the classical one-dimensional result (Canuto et al., 2007) to multivariate functions.

**Theorem E.2.** *Let $u : [-1,1]^d \to \mathbb{R}$, and let $u(\boldsymbol{x}) = \sum_{\boldsymbol{k}\in\mathcal{K}} \hat{u}_{\boldsymbol{k}} T_{\boldsymbol{k}}(\boldsymbol{x})$, where $\hat{u}_{\boldsymbol{k}}$ is the Chebyshev coefficient, $T_{\boldsymbol{k}}(\boldsymbol{x}) = \prod_{i=1}^d T_{k_i}(x_i)$, and $T_n(x)$ is the Chebyshev polynomial (first kind). For $n > 0$, if all mixed partial derivatives $\partial^\alpha u$ (with $\alpha = (\alpha_1, \alpha_2, \ldots, \alpha_d)$ a multi-index and $|\alpha| = \alpha_1 + \alpha_2 + \cdots + \alpha_d \leq n$) satisfy $\partial^\alpha u \cdot \prod_{j=1}^d \frac{1}{\sqrt{1-x_j^2}} \in L^1([-1,1]^d)$, then*

$$|\hat{u}_{\boldsymbol{k}}| \leq \frac{C}{\|\boldsymbol{k}\|_1^n}, \quad \forall \boldsymbol{k} \in \mathcal{K}^*, \tag{27}$$

*where $C$ is a constant that depends on $u$.*

*Furthermore, if $u$ is analytic,[9] then*

$$|\hat{u}_{\boldsymbol{k}}| \leq C\, \rho^{-\|\boldsymbol{k}\|_1}, \quad \forall \boldsymbol{k} \in \mathcal{K}, \tag{28}$$

*where $C, \rho$ are constants that depend on $u$.*

For the Chebyshev basis, Theorem E.2 reveals the relationship between decay rate and smoothness. Compared with Theorem E.1, one can find that both the Chebyshev and Fourier bases have algebraic decay of coefficients for finitely smooth functions and exponential decay of coefficients for analytic functions. However, the definitions of smoothness and analyticity are slightly different. Aside from the difference in domain, smoothness for the Fourier basis is defined in $L^1$

---

[8] Here analytic means real-analytic on $[0, 2\pi]^d$, i.e., $u$ admits a holomorphic extension to a complex strip $S_{\boldsymbol{\alpha}} = \{\boldsymbol{z} \in \mathbb{C}^d : |\Im z_j| < \alpha_j, j = 1, \ldots, d\}$ for some $\boldsymbol{\alpha} > \boldsymbol{0}$.

[9] Here analytic means real-analytic on $[-1, 1]^d$, i.e., $u$ admits a holomorphic extension to the interior of the Bernstein ellipse $E_{\boldsymbol{\rho}} = \left\{\boldsymbol{z} \in \mathbb{C}^d : z_j = \frac{1}{2}\left(\rho_j e^{i\theta_j} + \frac{1}{\rho_j e^{i\theta_j}}\right), \theta_j \in [0, 2\pi) \; j = 1, \ldots, d\right\}$, for some $\boldsymbol{\rho} \geq \boldsymbol{1}$.

space, while smoothness for the Chebyshev basis is defined in $L_w^1$ space, which is a weighted $L^1$ space with $w = \frac{1}{\sqrt{1-x^2}}$. Furthermore, their analytic functions require different complex domain (For Fourier, $S_\alpha$, for Chebyshev $E_\rho$).

*Remark* E.1. Compared with Theorem E.1, Theorem E.2 replaces the anisotropic decay rate by an isotropic rate. The isotropic rate is a relaxed condition of anisotropic rate by $n = \min\{\boldsymbol{n}\}$ and $\rho = \min\{\boldsymbol{\rho}\}$.

For general spectral bases, since an advantage of spectral bases is their ability to produce decaying expansion coefficients when representing sufficiently smooth functions, the feature of coefficient decay exists in every spectral basis, but the classes of functions for which this decay is pronounced and optimal differ across different bases. Recall the universal notation in Equation (3): $u_N(\boldsymbol{x}, t) = \sum_{\boldsymbol{k} \in \mathcal{K}_N} \hat{u}_{\boldsymbol{k}}(t) \phi_{\boldsymbol{k}}(\boldsymbol{x})$.

For finitely smooth functions, the algebraic decay depends on the smoothness defined on the corresponding orthogonal space (e.g., $L^1$ for Fourier, $L_w^1$ for Chebyshev) and the associated Sturm–Liouville operator

$$\mathcal{L}\phi_n = \lambda_n \phi_n, \tag{29}$$

where $\lambda_n$ increases monotonically with $n$ and $\lambda_n \to \infty$ as $n \to \infty$. Suppose that the function $f$ satisfies the corresponding smoothness condition with $\boldsymbol{n}$, then

$$|\hat{u}_{\boldsymbol{k}}| \le C \prod_{j=1}^{d} \lambda_{k_j}^{-n_j/2}, \quad \forall \boldsymbol{k} \in \mathcal{K}^*. \tag{30}$$

Herein, since $\lambda_n$ is monotonically increasing with $n$, the growth rate directly dictates the decay rate of the coefficients: faster growth of $\lambda_n$ leads to more rapid algebraic decay.

For analytic functions, the exponential decay depends primarily on whether the function's analytic domain matches (or extends into) the analytic domain of the spectral basis in complex domain. And the speed of exponential decay actually depends on both the size of the function's analytic domain and the shape of the spectral basis's analytic domain.

Additionally, if a function is infinitely smooth but not analytic, then the coefficient decay rate is super-algebraic.

# F. Proof of Embedding

This section first proves Theorem 3.1 and then discusses it.

*Proof.* First, suppose that $\boldsymbol{k}$ is arbitrary and that $\boldsymbol{k} \in \mathcal{K}_t$. Recall the definition of $\varepsilon(|\mathcal{X}|)$

$$\left| \int_\Omega u(\boldsymbol{x})\phi_{\boldsymbol{k}}(\boldsymbol{x})\mathrm{d}\boldsymbol{x} - \sum_{\boldsymbol{x} \in \mathcal{X}} \omega(\boldsymbol{x})u(\boldsymbol{x})\phi_{\boldsymbol{k}}(\boldsymbol{x}) \right| \le \varepsilon(|\mathcal{X}|). \tag{31}$$

Suppose $\boldsymbol{\phi}_{\boldsymbol{k}}^{\mathcal{X}} = \{\phi_{\boldsymbol{k}}(\boldsymbol{x})\}_{\boldsymbol{x} \in \mathcal{X}} \in \mathbb{R}^{|\mathcal{X}|}$ is the vector that every element is the corresponding discrete value of $\phi_{\boldsymbol{k}}$, $\boldsymbol{w}^{\mathcal{X}} = \{\omega(\boldsymbol{x})u(\boldsymbol{x})\}_{\boldsymbol{x} \in \mathcal{X}} \in \mathbb{R}^{1 \times |\mathcal{X}|}$, and $\boldsymbol{a}$ is the learnable vector. Then

$$\begin{aligned} \left| \hat{u}_{\boldsymbol{k}}^{\mathcal{X}} - \hat{u}_{\boldsymbol{k}}^{\mathcal{X}_c} \right| &= \left| \boldsymbol{w}^{\mathcal{X}} \cdot \boldsymbol{\phi}_{\boldsymbol{k}}^{\mathcal{X}} - \boldsymbol{w}^{\mathcal{X}_c} \cdot \boldsymbol{\phi}_{\boldsymbol{k}}^{\mathcal{X}_c} \right| \\ &\le \left| \int_\Omega u(\boldsymbol{x})\phi_{\boldsymbol{k}}(\boldsymbol{x})\mathrm{d}\boldsymbol{x} - \boldsymbol{w}^{\mathcal{X}} \cdot \boldsymbol{\phi}_{\boldsymbol{k}}^{\mathcal{X}} \right| + \left| \int_\Omega u(\boldsymbol{x})\phi_{\boldsymbol{k}}(\boldsymbol{x})\mathrm{d}\boldsymbol{x} - \boldsymbol{w}^{\mathcal{X}_c} \cdot \boldsymbol{\phi}_{\boldsymbol{k}}^{\mathcal{X}_c} \right| \\ &\le \varepsilon(|\mathcal{X}|) + \varepsilon(|\mathcal{X}_c|). \end{aligned} \tag{32}$$

The network $z^a(\boldsymbol{k})$ is designed to approximate $\hat{u}_{\boldsymbol{k}}^{\mathcal{X}}$. Thus, the parameter $\boldsymbol{a}$ is obtained by optimizing:

$$\min_a \left| z^a(\boldsymbol{k}) - \hat{u}_{\boldsymbol{k}}^{\mathcal{X}} \right| \Leftrightarrow \min_{\boldsymbol{a}} \left| \boldsymbol{a} \cdot \boldsymbol{\phi}_{\boldsymbol{k}}^{\mathcal{X}_c} - \boldsymbol{w}^{\mathcal{X}} \cdot \boldsymbol{\phi}_{\boldsymbol{k}}^{\mathcal{X}} \right|. \tag{33}$$

Since there exists $\boldsymbol{a}$ such that $\boldsymbol{a} = \boldsymbol{w}^{\mathcal{X}_c}$,

$$\min_{\boldsymbol{a}} \left| \boldsymbol{a} \cdot \boldsymbol{\phi}_{\boldsymbol{k}}^{\mathcal{X}_c} - \boldsymbol{w}^{\mathcal{X}} \cdot \boldsymbol{\phi}_{\boldsymbol{k}}^{\mathcal{X}} \right| = \left| \boldsymbol{w}^{\mathcal{X}} \cdot \boldsymbol{\phi}_{\boldsymbol{k}}^{\mathcal{X}} - \boldsymbol{w}^{\mathcal{X}_c} \cdot \boldsymbol{\phi}_{\boldsymbol{k}}^{\mathcal{X}_c} \right| \le \varepsilon(|\mathcal{X}|) + \varepsilon(|\mathcal{X}_c|) = \mathcal{O}(\varepsilon(|\mathcal{X}_c|)). \tag{34}$$

However, since the neural network is difficult to exactly satisfy the aforementioned condition. Thus, suppose that

$$a^* = \arg\min_{a} \left| a \cdot \phi_{\boldsymbol{k}}^{\mathcal{X}_c} - \boldsymbol{w}^{\mathcal{X}} \cdot \phi_{\boldsymbol{k}}^{\mathcal{X}} \right|, \tag{35}$$

Consequently, if the trained network satisfies $\left| a^* \cdot \phi_{\boldsymbol{k}}^{\mathcal{X}_c} - \boldsymbol{w}^{\mathcal{X}_c} \cdot \phi_{\boldsymbol{k}}^{\mathcal{X}_c} \right| \ll e\left( |\mathcal{X}_c| \right)$, then

$$\left| z^{a^*}(\boldsymbol{k}) - \hat{u}_{\boldsymbol{k}}^{\mathcal{X}} \right| = \left| a^* \cdot \phi_{\boldsymbol{k}}^{\mathcal{X}_c} - \boldsymbol{w}^{\mathcal{X}} \cdot \phi_{\boldsymbol{k}}^{\mathcal{X}} \right| = \mathcal{O}(\varepsilon(|\mathcal{X}_c|)). \tag{36}$$

However, since $\left| a^* \cdot \phi_{\boldsymbol{k}}^{\mathcal{X}_c} - \boldsymbol{w}^{\mathcal{X}_c} \cdot \phi_{\boldsymbol{k}}^{\mathcal{X}_c} \right| = 0 \Rightarrow a^* = \boldsymbol{w}^{\mathcal{X}_c}$ or $\left( a^* - \boldsymbol{w}^{\mathcal{X}} \right) \perp \phi_{\boldsymbol{k}}^{\mathcal{X}_c}$, Equation (36) cannot be extended to any $\boldsymbol{k} \in \mathcal{K}_w$.

Suppose $\Phi^{\mathcal{X}_c} = \left[ \phi_{\boldsymbol{k}}^{\mathcal{X}_c} \right]_{\boldsymbol{k} \in \mathcal{K}_t} \in \mathbb{R}^{|\mathcal{X}_c| \times |\mathcal{K}_t|}$ is the matrix constructed by vectors. Since $a^* = \boldsymbol{w}^{\mathcal{X}_c} \Leftrightarrow a^* \Phi^{\mathcal{X}_c} - \boldsymbol{w}^{\mathcal{X}_c} \Phi^{\mathcal{X}_c} = 0$ holds when $\mathrm{Rank}\left( \Phi^{\mathcal{X}_c} \right) \geq |\mathcal{X}_c|$. Thus, to satisfy that $\left\| a^* - \boldsymbol{w}^{\mathcal{X}_c} \right\|_2 \ll \frac{\varepsilon(|\mathcal{X}_c|)}{\|\phi_{\boldsymbol{k}}^{\mathcal{X}_c}\|_2}$, the training dataset $\mathcal{K}_t$ should guarantee that the constructed $\Phi^{\mathcal{X}_c}$ satisfies $\mathrm{Rank}\left( \Phi^{\mathcal{X}_c} \right) \geq |\mathcal{X}_c|$. Then

$$\left| a^* \cdot \phi_{\boldsymbol{k}}^{\mathcal{X}_c} - \boldsymbol{w}^{\mathcal{X}_c} \cdot \phi_{\boldsymbol{k}}^{\mathcal{X}_c} \right| \leq \left\| a^* - \boldsymbol{w}^{\mathcal{X}_c} \right\|_2 \|\phi_{\boldsymbol{k}}^{\mathcal{X}_c}\|_2 \ll e\left( |\mathcal{X}_c| \right), \ \forall \boldsymbol{k} \in \mathcal{K}_w. \tag{37}$$

Consequently, we can claim that

$$\left| z^{a^*}(\boldsymbol{k}) - \hat{u}_{\boldsymbol{k}}^{\mathcal{X}} \right| = \mathcal{O}(\varepsilon(|\mathcal{X}_c|)), \quad \forall \boldsymbol{k} \in \mathcal{K}_w \setminus \mathcal{K}_t. \tag{38}$$

$\square$

*Remark* F.1. Since $\phi_{\boldsymbol{k}}(\boldsymbol{x}) = \prod_{i=1}^{d} \phi_{k_i}(x_i)$, where $\boldsymbol{x} = \{x_i\}_{i=1}^{d}$ and $\boldsymbol{k} = \{k_i\}_{i=1}^{d}$. Thus, for high-dimensional problems in Modified SINNs, we first discretize every dimension to $N$ points, denoted by $\boldsymbol{x}_i = \{x_i^j\}_{j=1}^{N}$ for $i = 1, \cdots, d$. If $u$ is separable, then we can construct the matrix $\Phi = \left\{ \phi_{k_i}\left( x_i^j \right) \right\}_{i,j=1}^{d,N} \in \mathbb{R}^{d,N}$. Suppose $a \in \mathbb{R}^N$ is the simplified expression of learnable parameters that including weights $w$ and coefficients $c$. Let $b = \Phi a$ then every element $b_i$ of $b$ corresponds to the integration of $\phi_{k_i}(x_i)$. Finally, we output $z^a(\boldsymbol{k}) = \prod_{i=1}^{d} b_i$. If $u$ is not separable, we can also construct the matrix $\Phi = \left\{ \phi_{k_i}\left( x_i^j \right) \right\}_{i,j=1}^{d,N} \in \mathbb{R}^{d,N}$ with a position embedding. In this situation, $a$ is supposed not only learn weights $w$ and coefficients $c$, but also learn the integration of other dimensions. Furthermore, to avoid the vanish of production when $d$ is large, one can use $z^a(\boldsymbol{k}) = \sum_{i=1}^{d} b_i$.

*Remark* F.2. Lemma F.1 provides the sufficient condition to construct the $\Phi^{\mathcal{X}_c}$ with $\mathrm{Rank}\left( \Phi^{\mathcal{X}_c} \right) \geq |\mathcal{X}_c|$. Herein, in high-dimensional problems, $|\mathcal{K}_t| \geq |\mathcal{X}_c|$ is enough to guarantee $\mathrm{Rank}\left( \Phi^{\mathcal{X}_c} \right) \geq |\mathcal{X}_c|$, especially the sampling distribution based on hyperbolic cross makes the sampled $\mathcal{K}_t$ concentrate on the low-frequency part.

**Lemma F.1.** *For arbitrary* $\boldsymbol{k_1} \neq \boldsymbol{k_2}$, $\boldsymbol{k_1} = \{k_{1,i}\}_{i=1}^{d} \in \mathcal{K}_w$, $\boldsymbol{k_2} = \{k_{2,i}\}_{i=1}^{d} \in \mathcal{K}_w$,

$$\phi_{\boldsymbol{k_1}} \cdot \phi_{\boldsymbol{k_2}} \neq 0 \Leftrightarrow \quad k_{1,i} \equiv k_{2,i} \,(mod\ N) \quad \forall i = 1, \cdots d,$$

*where* $N$ *is a constant that depends on* $|\mathcal{X}_c|$ *and* $\phi$.

## G. Convergence Analysis for SINNs in High-dimensional PDEs

Since the additional blocks can be regarded as multiplicative reparameterizations on $\hat{u}^{\theta}$ and are uniformly bounded, Modified SINNs do not violate the error convergence theorem for SINNs in (Yu et al., 2025). However, the error convergence theorem in (Yu et al., 2025) is based on the assumption of one-dimensional cases. Here, we extend the error convergence theorem to high-dimensional problems.

Same as (Yu et al., 2025), since temporal error is much smaller than spatial error, we ignore the temporal error in this analysis. For SINNs, suppose the analytic solution is $u^*(\boldsymbol{x})$ and the truncated expansion of the solution is $u_N^*(\boldsymbol{x}) = \sum_{\boldsymbol{k} \in \mathcal{K}_N} \hat{u}_{\boldsymbol{k}}^* \phi_{\boldsymbol{k}}(\boldsymbol{x})$. Let $\hat{u}^{\theta}(\boldsymbol{k}, t)$ be the output of SINNs and $u^{\theta}(\boldsymbol{x}) = \sum_{\boldsymbol{k} \in \mathcal{K}_N} \hat{u}^{\theta}(\boldsymbol{k}, t) \phi_{\boldsymbol{k}}(\boldsymbol{x})$. Then, define $\theta_N^*$ is the parameters after training with $\mathcal{K}_N$:

$$\theta_N^* := \arg\min_{\theta} \sum_{\boldsymbol{k} \in \mathcal{K}_N} \mathcal{L}\left[ u^{\theta}(\boldsymbol{k}) \right]_{\Omega}, \tag{39}$$

Then, we have

$$\left\| u^*(\boldsymbol{x}) - \hat{u}^{\theta_N^*}(\boldsymbol{x}) \right\|_\Omega \leq \left\| u^*(\boldsymbol{x}) - u_N^*(\boldsymbol{x}) \right\|_\Omega + \left\| u_N^*(\boldsymbol{x}) - u^{\theta_N^*}(\boldsymbol{x}) \right\|_\Omega. \tag{40}$$

Based on universal approximation theorem (Hornik et al., 1989), the second term can be made arbitrarily small with sufficiently large network capacity. Thus, the error convergence of SINNs is determined by the first term, which is the error of spectral expansion. Since the spectral expansion depends on the methods for generating the index set $\mathcal{K}_N$, the error convergence of SINNs is determined by the method for generating $\mathcal{K}_N$. If the index set $\mathcal{K}_N$ is generated by methods which have spectral convergence, the error convergence of SINNs is also spectral.

## H. Derivatives of Chebyshev Polynomials

There are many accurate and efficient methods for computing derivatives in Chebyshev spectral methods. However, since this paper focuses on sparse coefficients and avoids transforming them back to the physical domain frequently, we introduce an approach that computes derivatives directly in the spectral domain. Notably, although previous works (Liu et al., 2011; Oh, 2019) have derived schemes for the derivatives of Chebyshev polynomials in the spectral domain, these studies primarily address dense coefficients and low-dimensional PDEs.

Suppose that $D \in \mathbb{R}^{N_x \times N_x}$ is the Chebyshev differential matrix, the function $u(x) = \sum_{k=0}^{N_x-1} c_k T_k(x) \in L_w^2[-1, 1]$, and $\boldsymbol{c} = \{c_i\}_{i=0}^{N_x-1}$. If $\{x_i\}_{i=1}^{N_x}$ is the Chebyshev–Lobatto nodes, suppose $\boldsymbol{u} = \{u(x_i)\}_{i=1}^{N_x}$, and $T = \{t_{ij}\} \in \mathbb{R}^{N_x \times N_x}$ where $t_{ij} = T_i(x_j)$. Then

$$\begin{aligned} \boldsymbol{u} &= T\boldsymbol{c}, \\ \boldsymbol{u}' &= DT\boldsymbol{c}, \\ &\vdots \\ \boldsymbol{u}^{(n)} &= D^n T\boldsymbol{c}, \end{aligned} \tag{41}$$

Based on Chebyshev–Gauss quadrature, one can obtain that, for $B = \frac{2}{N_x} T^T \text{diag}\left(\{w_i\}_{i=0}^{N_x-1}\right)$ where $w_0 = w_{N_x-1} = 1/2$, $w_i = 1$, $\forall i = 1, \cdots, N_x - 2$, $TB = I$ where $I$ is identity matrix. Suppose $D_i = BD^i T$, then

$$\begin{aligned} TD_1\boldsymbol{c} &= TBDT\boldsymbol{c} = DT\boldsymbol{c} = \boldsymbol{u}', \\ &\vdots \\ TD_n\boldsymbol{c} &= TBD^n T\boldsymbol{c} = D^n T\boldsymbol{c} = \boldsymbol{u}^{(n)}. \end{aligned} \tag{42}$$

If $u^{(n)} \in L_w^2[-1, 1]$, by Lemma H.1, $\boldsymbol{c}_n = D_n\boldsymbol{c}$ is the vector of Chebyshev coefficients of the $n$th-order derivative of $u$. Then for multi-variable function $u(\boldsymbol{x})$, suppose the corresponding function tensor is $\mathcal{U}$ and the coefficient tensor is $\mathcal{C}$, then

$$\partial_\alpha^n \mathcal{C} = \mathcal{C} \times_\alpha D_n, \tag{43}$$

where $\partial_\alpha^n$ is the $n$-th derivative of the $\alpha$-th variable and $\times_\alpha$ is the contraction operation along the $\alpha$-th dimension.

**Lemma H.1.** *(Yosida, 1980) Let $(L_w^2([-1, 1]), \langle \cdot, \cdot \rangle_w)$ be the weighted $L^2$–space with weight $w(x) > 0$ a.e. on $[-1, 1]$, and let $\{\phi_i\}_{i \in I}$ be an orthonormal basis of $L_w^2([-1, 1])$. Then for every $u \in L_w^2([-1, 1])$, the coefficients $\alpha_i = \langle u, \phi_i \rangle_w$, $i \in I$, are uniquely determined.*

Furthermore, by tensor operation laws:

$$\begin{aligned} \partial_\alpha^n \mathcal{C} &= \mathcal{C} \times_\alpha D_n \\ &\Leftrightarrow \text{vec}\left(\partial_\alpha^n \mathcal{C}\right) = M_{D_n}^\alpha \text{vec}(\mathcal{C}), \\ \text{where} \quad M_{D_n}^\alpha &= \underbrace{I \otimes \cdots \otimes I}_{\alpha-1} \otimes D_n \otimes \underbrace{I \otimes \cdots \otimes I}_{d-\alpha} \quad \text{is a matrix.} \end{aligned} \tag{44}$$

Thus, constructing the matrix is the key operation in high-dimensional Chebyshev spectral methods.

Notably, to obtain a more accurate $D_i$, one can use $D_i = BD^{(i)} T$, where $D^{(i)}$ is the $i$-th order differential matrix generated directly.

**H.1. Construction of Sparse Differentiation Matrix.**

Obviously, if $\mathcal{C}$ is a $d$-dimensional tensor, then the complexity of a single dimension contraction operation is $\mathcal{O}(N_x^{d+1})$. Thus to avoid the CoD, we derive the sparse differentiation matrix for sparse coefficients, *i.e.*, the matrix $M_D$, similar to the notation in Equation (44).

Let $\mathcal{K} = \{\boldsymbol{k_i}\}$ be a sparse multi-index set with $\#\mathcal{K} = N_{\text{valid}}$, where $\boldsymbol{k_i} = (k_{i_1}, \ldots, k_{i_d})$. We consider the tensor-product Chebyshev expansion

$$u(\boldsymbol{x}) = \sum_{\boldsymbol{k_i} \in \mathcal{K}} c_{\boldsymbol{k_i}} \prod_{j=1}^d T_{k_{i_j}}(x_{i_j}) := T\boldsymbol{c}. \tag{45}$$

Since $D \in \mathbb{R}^{N_x \times N_x}$ is the one-dimensional Chebyshev differentiation matrix in coefficient space. For differentiation with respect to the $\alpha$-th dimension, ($\alpha \in \{1, \ldots, d\}$), we construct the sparse spectral differentiation matrix $M_\alpha = \{m_{ij}^\alpha\} \in \mathbb{R}^{N_{\text{valid}} \times N_{\text{valid}}}$ by

$$m_{ij}^\alpha = \begin{cases} D_{k_{i_\alpha}, \, k_{j_\alpha}}, & \text{if } k_{i_\ell} = k_{j_\ell} \text{ for all } \ell \neq \alpha, \\ 0, & \text{otherwise.} \end{cases} \tag{46}$$

This construction corresponds to restricting the tensor-product operator $I \otimes \cdots \otimes D_c \otimes \cdots \otimes I$ to the sparse index set $\mathcal{K}$. Consequently, suppose the Chebyshev coefficients of the partial derivative $\partial_\alpha^{(n)} u$ is $\partial_\alpha^{(n)} \boldsymbol{c} \in \mathbb{R}^{N_{valid}}$, then $\partial_\alpha^{(n)} \boldsymbol{c}$ is given by

$$\partial_\alpha^{(n)} \boldsymbol{c} = M_\alpha^n \boldsymbol{c}. \tag{47}$$

Thus,

$$\partial_\alpha^{(n)} u = T M_\alpha^n \boldsymbol{c}. \tag{48}$$

Since the construction of $M_\alpha$ does not compute the full tensor operator but directly computes valid sparse index mappings, it is computationally efficient. The construction cost of $M_\alpha$ is $\mathcal{O}(N_x^3 + N_x |\mathcal{K}|)$. This complexity is not explicitly exponential in $d$; instead, it is controlled by $|\mathcal{K}|$. In other words, regardless of how large $d$ is, the construction remains efficient as long as the spectral index set is sparse.

# I. Experimental Details for Approximating Missing Coefficients

In this section, we provide the full experimental results for approximating missing coefficients with different activation functions and sizes of MLP in Section 4.1. Then we provide a demonstration in Appendix I.1 to show the performance of SINNs in approximating missing coefficients for complex functions. In this experiment, we first generate $\hat{f}_k = e^{|k|} a_k$, $a_k \sim \mathcal{N}(0, 1)$, for $k \neq 0$ and $\hat{f}_0 = 1$ and then $\hat{f}_{\boldsymbol{k}} = \prod_{i=1}^d \hat{f}_{k_i}$. To avoid CoD, we only calculate and use the valid $\hat{f}_{\boldsymbol{k}}$ and their corresponding index set $\mathcal{K}$. Therefore, given the input dataset $\mathcal{K}$, the corresponding output can be represented as the set of input-output pairs $\left\{ (\boldsymbol{k}, \hat{f}_{\boldsymbol{k}}) | \boldsymbol{k} \in \mathcal{K} \right\}$.

First, the experiments with $s = 0.9$ are provided in Table 6 for Fourier and Table 7 for Chebyshev with different activation functions and sizes of MLP. For SINNs with Fourier basis, for $d = 2$, the $5 \times 100$ MLP with the GELU activation function minimizes the error; for $d = 3$, the $10 \times 50$ MLP with the tanh activation function is optimal; for $d = 6$, the $5 \times 100$ MLP with the tanh activation function achieves the optimal error; and for $d = 8$, the $5 \times 50$ MLP with the tanh activation function yields the lowest error. For SINNs with Chebyshev basis, for $d = 2$, the $10 \times 100$ MLP with the GELU activation function minimizes the error; for $d = 3$, the $5 \times 50$ MLP with tanh activation is optimal; for $d = 5$, the $10 \times 50$ MLP and tanh activation yield the lowest error; and for $d = 6$, the $5 \times 50$ MLP with tanh activation achieves the optimal error. We also provide the training dynamics of the optimal models in Figure 9 for Fourier and Figure 8 for Chebyshev. Additionally, we provide a non-separable benchmark with the solution $u(\boldsymbol{x}) = \left( \sum_{i=1}^d \frac{1}{d} x_i \right)^2 + \sin\left( \sum_{i=1}^d \frac{1}{d} x_i \right)$, $\boldsymbol{x} \in [-\pi, \pi]^d$ in Table 8 to further evaluate the performance of SINNs. We also provide a comparison for approximating the missing coefficients that are sampled from a distribution with higher density over high-frequency components in Table 9.

In general, without any prior knowledge of either the function or the unknown coefficients, knowing 90% of the coefficients in the spectral domain makes it impossible to determine the remaining 10%. However, the potential low-dimensional

structure of physical functions in the spectral domain makes the reconstruction possible and feasible. For example, in spectral methods, with prior knowledge of sparsity, compressed sensing can predict the whole coefficient matrix from randomly sampled frequencies with a much smaller number of samples (Tropp & Gilbert, 2007; Figueiredo et al., 2008); spectral extrapolation can recover unobserved frequency components from a subset of known spectral coefficients (Schmidt, 1986; Candès & Fernandez-Granda, 2014). Uncovering low-dimensional structures embedded in complex data is a core capability of machine learning. Consequently, SINNs are expected to leverage this capability to approximate such structures and thereby predict missing information. Furthermore, the modified parts that integrate prior information enhance approximation accuracy.

*Table 6.* Predicting missing coefficients for Fourier with $s = 0.9$.

| DIM | SIZE | TANH | GELU | SILU | $|\mathcal{X}_c|$ |
|---|---|---|---|---|---|
| 2 | $5 \times 50$ | 6.21E-4 ± 4.00E-6 | 6.32E-4 ± 2.73E-4 | 4.58E-4 ± 1.18E-4 | 8 |
| | $5 \times 100$ | 6.11E-4 ± 2.60E-5 | 4.21E-4 ± 8.00E-5 | 4.62E-4 ± 9.70E-5 | 8 |
| | $10 \times 50$ | 6.28E-4 ± 1.90E-5 | 4.98E-4 ± 1.67E-4 | 6.33E-4 ± 2.30E-5 | 8 |
| | $10 \times 100$ | 6.26E-4 ± 1.80E-5 | 6.40E-4 ± 2.00E-6 | 6.41E-4 ± 4.00E-6 | 8 |
| 3 | $5 \times 50$ | 1.10E-4 ± 2.70E-5 | 1.34E-4 ± 1.90E-5 | 1.04E-4 ± 3.10E-5 | 12 |
| | $5 \times 100$ | 1.52E-4 ± 4.70E-5 | 1.60E-4 ± 9.00E-6 | 1.57E-4 ± 3.90E-5 | 12 |
| | $10 \times 50$ | 5.60E-5 ± 1.40E-5 | 9.17E-3 ± 1.29E-2 | 3.33E-1 ± 4.71E-1 | 12 |
| | $10 \times 100$ | 1.13E-4 ± 4.10E-5 | 1.48E-4 ± 1.40E-5 | 1.38E-4 ± 4.20E-5 | 12 |
| 6 | $5 \times 50$ | > 1 | > 1 | 6.68E-1 ± 4.70E-1 | 24 |
| | $5 \times 100$ | 2.40E-2 ± 3.23E-2 | 6.67E-1 ± 4.71E-1 | 3.78E-1 ± 4.43E-1 | 24 |
| | $10 \times 50$ | 6.67E-1 ± 4.71E-1 | 6.67E-1 ± 4.71E-1 | 6.67E-1 ± 4.71E-1 | 24 |
| | $10 \times 100$ | 7.04E-1 ± 4.19E-1 | > 1 | > 1 | 24 |
| 8 | $5 \times 50$ | 3.92E-3 ± 2.42E-4 | 3.36E-1 ± 4.69E-1 | 6.68E-1 ± 4.69E-1 | 32 |
| | $5 \times 100$ | > 1 | > 1 | > 1 | 32 |
| | $10 \times 50$ | > 1 | 6.67E-1 ± 4.71E-1 | > 1 | 32 |
| | $10 \times 100$ | 7.83E-1 ± 3.07E-1 | > 1 | > 1 | 32 |

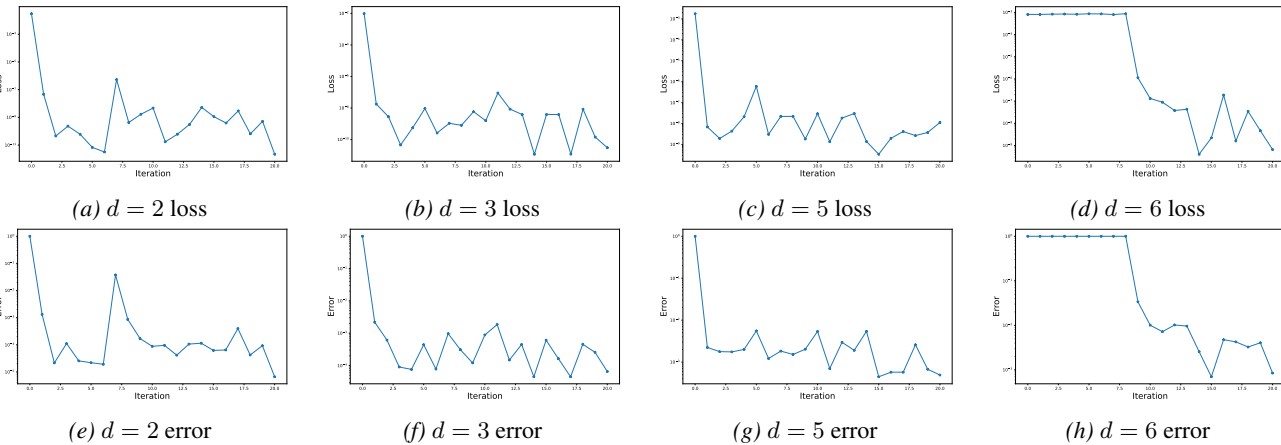

*(a) d = 2 loss*  *(b) d = 3 loss*  *(c) d = 5 loss*  *(d) d = 6 loss*

*(e) d = 2 error*  *(f) d = 3 error*  *(g) d = 5 error*  *(h) d = 6 error*

*Figure 8.* Training dynamics for Chebyshev approximation. The first row shows training loss and the second row shows relative errors of $u$.

## I.1. Complex Function Demonstration

In this section, the SINN is trained to approximate the Chebyshev coefficient matrix $C = \{c_{ij}\}_{i,j=1}^{50}$ corresponding to the function $u(x,y) = e^{-5(x^2+y^2)} \sin(5x) \sin(3y) + 0.2 \cos(10xy)$, $x, y \in [0,1]$ where only 90% of the coefficients are randomly sampled from the matrix $C$ and used for training. We depict the performance in Figure 3 by the full distribution of

*Table 7.* Predicting missing coefficients for Chebyshev with $s = 0.9$.

| DIM | SIZE | TANH | GELU | SILU | $|\mathcal{X}_c|$ |
|---|---|---|---|---|---|
| 2 | $5 \times 50$ | 5.79E-4 $\pm$ 4.01E-4 | 5.87E-4 $\pm$ 3.60E-4 | 4.79E-4 $\pm$ 2.65E-4 | 8 |
| | $5 \times 100$ | 2.96E-4 $\pm$ 3.80E-4 | 1.80E-3 $\pm$ 2.23E-3 | 1.35E-3 $\pm$ 1.54E-3 | 8 |
| | $10 \times 50$ | 7.14E-4 $\pm$ 4.60E-5 | 1.05E-3 $\pm$ 1.43E-3 | 1.91E-4 $\pm$ 1.27E-4 | 8 |
| | $10 \times 100$ | 6.57E-4 $\pm$ 3.90E-5 | 5.50E-5 $\pm$ 5.30E-5 | 5.70E-5 $\pm$ 6.20E-5 | 8 |
| 3 | $5 \times 50$ | 3.60E-5 $\pm$ 1.20E-5 | 1.28E-2 $\pm$ 1.75E-2 | 2.25E-3 $\pm$ 2.30E-3 | 12 |
| | $5 \times 100$ | 2.47E-4 $\pm$ 1.61E-4 | 3.04E-2 $\pm$ 4.09E-2 | 6.45E-4 $\pm$ 4.27E-4 | 12 |
| | $10 \times 50$ | 3.77E-4 $\pm$ 8.90E-5 | 3.33E-1 $\pm$ 4.71E-1 | 1.59E-4 $\pm$ 6.90E-5 | 12 |
| | $10 \times 100$ | 3.34E-1 $\pm$ 4.71E-1 | 3.34E-1 $\pm$ 4.71E-1 | 3.33E-1 $\pm$ 4.71E-1 | 12 |
| 5 | $5 \times 50$ | 3.33E-1 $\pm$ 4.71E-1 | 3.26E-4 $\pm$ 9.10E-5 | 6.03E-4 $\pm$ 2.29E-4 | 20 |
| | $5 \times 100$ | 1.03E-1 $\pm$ 1.42E-1 | 6.68E-1 $\pm$ 4.69E-1 | 4.20E-1 $\pm$ 4.23E-1 | 20 |
| | $10 \times 50$ | 3.92E-4 $\pm$ 5.50E-5 | 4.35E-4 $\pm$ 1.40E-4 | 3.12E-2 $\pm$ 4.31E-2 | 20 |
| | $10 \times 100$ | 1.95E-1 $\pm$ 2.71E-1 | 6.68E-1 $\pm$ 4.63E-1 | 9.96E-1 $\pm$ 1.18E-3 | 20 |
| 6 | $5 \times 50$ | 8.88E-3 $\pm$ 0.00E-2 | $> 1$ | 3.42E-1 $\pm$ 4.66E-1 | 72 |
| | $5 \times 100$ | 3.61E-1 $\pm$ 4.51E-1 | 2.09E-1 $\pm$ 1.88E-1 | 6.51E-1 $\pm$ 4.56E-1 | 72 |
| | $10 \times 50$ | 6.74E-1 $\pm$ 4.61E-1 | 6.75E-1 $\pm$ 4.60E-1 | $> 1$ | 72 |
| | $10 \times 100$ | 4.05E-1 $\pm$ 4.21E-1 | 4.33E-1 $\pm$ 4.15E-1 | 3.79E-1 $\pm$ 4.41E-1 | 72 |

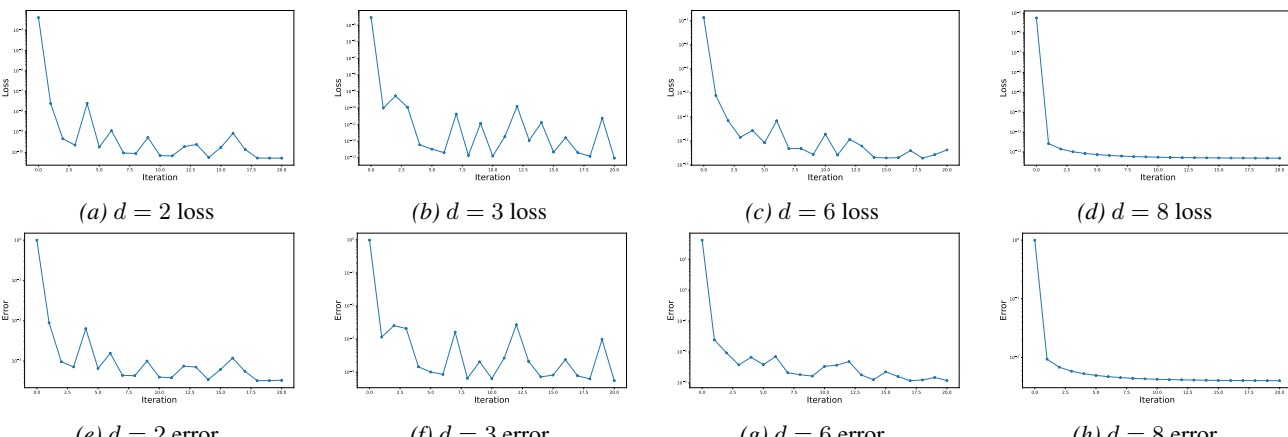

*(a) $d = 2$ loss*  *(b) $d = 3$ loss*  *(c) $d = 6$ loss*  *(d) $d = 8$ loss*

*(e) $d = 2$ error*  *(f) $d = 3$ error*  *(g) $d = 6$ error*  *(h) $d = 8$ error*

*Figure 9.* Training dynamics for Fourier approximation. The first row shows training loss and the second row shows relative errors of $u$.

$C$ and the masked distribution of $C$. Figure 3 reveals that the approximation error primarily stems from discrepancies in the magnitude of the coefficients, whereas the errors associated with the masked elements can be approximated with high accuracy.

From the basis embedding perspective, since the Chebyshev spectral method employs Chebyshev-Gauss-Lobatto quadrature, based on Lemma I.1, it follows that in Theorem 3.1, $\varepsilon(|\mathcal{X}|) = \frac{C^2}{R^{2|\mathcal{X}|}}$. Consequently, the error $|z^a(\boldsymbol{k}) - \hat{u}_{\boldsymbol{k}}^{\mathcal{X}}|$ converges exponentially. Herein, in 2D Chebyshev SINNs, the basis embedding module is feasible and efficient to accurately approximate $\hat{u}_{\boldsymbol{k}}^{\mathcal{X}}$.

**Lemma I.1.** *(Gautschi & Varga, 1983) If $f : \mathbb{R} \to \mathbb{R}$ is analytic, then the error of Chebyshev-Gauss quadrature:*

$$\left| \int_{-1}^{1} f(x) dx - \sum_{k=0}^{n-1} \omega(x_k) f(x_k) \right| \leq \frac{C}{R^n}. \tag{49}$$

*where $C, R$ are constant and depend on $f$.*

*Table 8.* Prediction missing errors for sine functions.

| DIM | $s$ | ERROR | ERROR (MISSING) | $|\mathcal{X}_c|$ |
|---|---|---|---|---|
| 2 | 1.0 | 3.41E-4 ± 4.40E-5 | – | 900 |
| | 0.9 | 6.82E-2 ± 1.13E-2 | 5.62E-1 ± 1.47E-1 | 900 |
| | 0.8 | 1.67E-1 ± 2.65E-2 | 6.24E-1 ± 1.20E-1 | 900 |
| 3 | 1.0 | 4.46E-3 ± 9.76E-4 | – | 27000 |
| | 0.9 | 4.99E-2 ± 1.95E-2 | 2.34E-1 ± 1.06E-1 | 27000 |
| | 0.8 | 7.72E-2 ± 1.34E-2 | 3.29E-1 ± 5.52E-2 | 27000 |
| 5 | 1.0 | 6.93E-4 ± 1.47E-4 | – | 7776 |
| | 0.9 | 4.76E-2 ± 5.13E-2 | 1.80E-1 ± 1.92E-1 | 7776 |
| | 0.8 | 8.97E-2 ± 1.14E-1 | 2.43E-1 ± 3.09E-1 | 7776 |
| 6 | 1.0 | 2.54E-3 ± 4.25E-4 | – | 46656 |
| | 0.9 | 7.54E-3 ± 4.48E-3 | 2.56E-2 ± 1.55E-2 | 46656 |
| | 0.8 | 2.04E-2 ± 1.30E-2 | 5.49E-2 ± 3.82E-2 | 46656 |

*Table 9.* Prediction missing errors with sampling from a frequency-based distribution for sine funcions.

| DIM | $s$ | ERROR | ERROR (MISSING) | $|\mathcal{X}_c|$ |
|---|---|---|---|---|
| 2 | 1.0 | 6.98E-4 ± 6.52E-5 | – | 900 |
| | 0.9 | 1.07E-1 ± 6.76E-2 | 5.89E-1 ± 1.12E-1 | 900 |
| | 0.8 | 1.86E-1 ± 1.45E-1 | 6.52E-1 ± 2.78E-1 | 900 |
| 3 | 1.0 | 5.43E-3 ± 5.21E-4 | – | 27000 |
| | 0.9 | 1.99E-2 ± 1.48E-2 | 1.55E-1 ± 9.99E-2 | 27000 |
| | 0.8 | 5.74E-2 ± 3.65E-2 | 4.47E-1 ± 3.64E-1 | 27000 |
| 5 | 1.0 | 1.01E-3 ± 2.16E-4 | – | 7776 |
| | 0.9 | 4.08E-2 ± 3.33E-2 | 1.34E-1 ± 1.04E-1 | 7776 |
| | 0.8 | 4.41E-2 ± 2.28E-2 | 1.10E-1 ± 5.70E-2 | 7776 |
| 6 | 1.0 | 2.00E-1 ± 3.42E-1 | – | 46656 |
| | 0.9 | 7.19E-3 ± 3.01E-3 | 2.99E-2 ± 1.48E-2 | 46656 |
| | 0.8 | 1.24E-2 ± 7.02E-3 | 3.30E-2 ± 1.88E-2 | 46656 |

## J. PDEs for Experiments

This section provides details of PDEs used in the experiments presented in Section 4. Specifically, Appendix J.1 describes the middle-dimensional PDEs employed in Section 4.2. The high-dimensional Poisson problems used to compare SGSM and SINNs in Section 4.2 are detailed in Appendix J.2. Finally, Appendix J.3 presents the time-dependent Schrödinger equations considered in the hybrid SGSM-SINN experiments of Section 4.4.

Across all experiments, the spectral coefficients used as ground truth are constructed either by (i) generating analytical coefficients from a normal distribution, (ii) obtaining coefficients along a selected subset of coordinate directions, or (iii) inducing them via separable functions. We argue that, under limited computational resources, high-dimensional spectral transforms inevitably introduce non-negligible numerical errors in the coefficients. Such errors make comparisons between different methods unfair. However, since real problems are usually given in physical space, further discussion of this issue is provided in Appendix J.4. Furthermore, to avoid CoD, we only calculate and use the valid $\hat{f}_{\boldsymbol{k}}$ and their corresponding index set $\mathcal{K}$.

### J.1. Middle-dimensional Equations

The experiments for middle-dimensional PDEs are all generated from analytical coefficients which are directly generated by random sampling. Since the functions used in middle-dimensional PDEs are real function, after generating the Fourier

coefficients $\hat{u}_{\boldsymbol{k}}$, we set $\hat{u}_{-\boldsymbol{k}} = \hat{u}_{\boldsymbol{k}}$ to guarantee the reality of the function.

### J.1.1. STEADY PDEs WITH ANALYTIC COEFFICIENTS

For steady PDEs, the only function that needs to be constructed is the right-hand-side term, as periodic boundary conditions are used for Poisson equations and Dirichlet boundary conditions are used for steady convection equations.

**Poisson's equations.** We consider the Poisson's equation with periodic boundary condition:

$$\Delta u(\boldsymbol{x}) = f(\boldsymbol{x}), \quad \boldsymbol{x} \in [0, 2\pi]^d. \tag{50}$$

We construct $f(\boldsymbol{x}) = \prod_{i=1}^{d} \tilde{f}(x_i)$, $\boldsymbol{x} = (x_1, \cdots, x_d) \in \mathbb{R}^d$, where $\tilde{f}(x) = \sum_k \operatorname{Re}\left(\hat{f}_k e^{\mathrm{i}kx}\right)$, $\hat{f}_k \in \mathbb{R}$. Suppose $\boldsymbol{k} = (k_1, \cdots, k_d)$ is the Fourier index vector in the $d$ dimensional spectral space, the discrete multi-dimensional Fourier coefficient can be expressed exactly $\hat{f}_{\boldsymbol{k}} = \prod_{i=1}^{d} \hat{f}_{k_i}$. Consequently, the solution $u(\boldsymbol{x}) = \sum_{\boldsymbol{k}} \operatorname{Re}\left(\hat{u}_{\boldsymbol{k}} e^{\mathrm{i}\boldsymbol{k}\cdot\boldsymbol{x}}\right)$, where $\hat{u}_{\boldsymbol{k}} = -\hat{f}_{\boldsymbol{k}}/\|\boldsymbol{k}\|_2^2$. We randomly generate $\hat{f}_{k_i} = e^{-|k|}a_{k_i}$, $a_{k_i} \sim \mathcal{N}(0,1)$, for $k_i \neq 0$ and $\hat{f}_0 = 0$, to guarantee the coefficient decay condition (analytic function) and avoid the singularity of $\hat{u}_{\boldsymbol{k}}$.

**Convection equations.** Here we consider the convection equations with Dirichlet boundary condition:

$$\sum_{i=1}^{d} u_{x_i}(\boldsymbol{x}) = f(\boldsymbol{x}), \quad \boldsymbol{x} \in [-1, 1]^d. \tag{51}$$

where $f(\boldsymbol{x}) = \prod_{i=1}^{d} \tilde{f}(x_i)$, and $\tilde{f}(x) = \sum_{k \in \mathcal{K}} \hat{f}_k T_k(x)$, $\hat{f} \in \mathbb{R}$. We randomly generate $\hat{f}_k = e^{-|k|}a_k$, $a_k \sim \mathcal{N}(0,1)$, for $k \neq 0$ to guarantee the coefficient decay condition (analytic function). we also set $\hat{f}_0 = 0$ since it represents the constant term. To avoid CoD, we only calculate and use the valid $\hat{f}_{\boldsymbol{k}}$ and their corresponding index set $\mathcal{K}$. Suppose $\boldsymbol{c}_f$ is the vector of $\hat{f}_k$ for $k \in \mathcal{K}$, $\boldsymbol{c}_u$ is the vector of $\hat{u}_k$ for $k \in \mathcal{K}$, and the differential matrix $M$ is constructed by Appendix H, then we can get $c_u = M^{-1}c_f$, and then calculate $u(\boldsymbol{x}) = \sum_{k \in \mathcal{K}} \hat{u}_k T_{\boldsymbol{k}}(\boldsymbol{x})$. Thus the Dirichlet boundary condition is

$$u(\boldsymbol{x}) = \sum_{k \in \mathcal{K}} \hat{u}_k T_{\boldsymbol{k}}(\boldsymbol{x}), \quad \boldsymbol{x} \in \partial[-1, 1]^d. \tag{52}$$

### J.1.2. TIME-DEPENDENT PDEs WITH ANALYTIC COEFFICIENTS

**Heat equations.** Consider the heat equation with periodic boundary condition:

$$u_t(\boldsymbol{x}, t) = \Delta u(\boldsymbol{x}, t), \quad (\boldsymbol{x}, t) \in [0, 2\pi]^d \times [0, 0.1]. \tag{53}$$

Here we consider a separable function $u(\boldsymbol{x}, 0) = \prod_{i=1}^{d} g(x_i)$ as the initial condition, where $g(x) = \sum_k \operatorname{Re}\left(\hat{u}_{k,0} e^{\mathrm{i}kx}\right)$, $\hat{u}_{k,0} \in \mathbb{C}$. Then the Fourier coefficient can be exactly expressed by $\hat{u}_{\boldsymbol{k},0} = 2^{-d} \prod_{i=1}^{d} \hat{u}_{k_i,0}$. Consequently, the solution $u(\boldsymbol{x}, t) = \sum_{\boldsymbol{k}} \operatorname{Re}\left(\hat{u}_{\boldsymbol{k},t} e^{\mathrm{i}\boldsymbol{k}\cdot\boldsymbol{x}}\right)$, where $\hat{u}_{\boldsymbol{k},t} = \prod_{i=1}^{d} \hat{u}_{k_i,0} e^{-\|\boldsymbol{k}\|_2^2 t}$. We randomly generate $\hat{u}_{k,0} = e^{-|k|}(a_k + b_k \mathrm{i})$, $a_k, b_k \sim \mathcal{N}(0,1)$ for $k \neq 0$ and $\hat{u}_{0,0} = 0$ to avoid the singularity.

**Convection equations.** Consider the convection equations with Dirichlet boundary condition:

$$\sum_{i=1}^{d} u_{x_i}(\boldsymbol{x}, t) = u_t(\boldsymbol{x}, t), \quad (\boldsymbol{x}, t) \in [-1, 1]^d \times [0, 0.1]. \tag{54}$$

The initial condition is $u(\boldsymbol{x}, 0) = \prod_{i=1}^{d} g(x_i)$, where $g(x) = \sum_k \hat{u}_{k,0} T_k(x)$, $\hat{u}_{k,0} \in \mathbb{R}$. Consequently, the solution $u(\boldsymbol{x}, t) = \prod_{i=1}^{d} g(x_i + t)$. We randomly generate $\hat{u}_{k,0} = e^{-|k|}a_k$, $a_k \sim \mathcal{N}(0,1)$, for $k \neq 0$ to guarantee the coefficient decay condition (analytic function). we also set $\hat{f}_0 = 0$ since it represents the constant term.

*Table 10.* PINNs and DRMs on middle-dimensional PDEs.

| EQUATION | DIM | PINN | DRM |
|---|---|---|---|
| POISSON | 2 | $1.10\text{E-}3 \pm 2.08\text{E-}5$ | $3.47\text{E-}3 \pm 1.76\text{E-}4$ |
| | 3 | $1.18\text{E-}2 \pm 2.70\text{E-}3$ | $9.82\text{E-}3 \pm 6.52\text{E-}4$ |
| | 6 | $7.30\text{E-}1 \pm 2.91\text{E-}1$ | $1.13\text{E-}1 \pm 2.38\text{E-}2$ |
| | 8 | $> 1$ | $3.99\text{E-}1 \pm 1.59\text{E-}2$ |
| HEAT | 2 | $2.78\text{E-}2 \pm 1.08\text{E-}2$ | $2.54\text{E-}3 \pm 1.35\text{E-}4$ |
| | 3 | $3.80\text{E-}2 \pm 1.50\text{E-}3$ | $4.53\text{E-}3 \pm 1.46\text{E-}4$ |
| | 6 | $> 1$ | $> 1$ |

## J.2. High-dimensional Equations

**Poisson's equations.** For the Poisson's equation considered in Section 4.2, we first define the right-hand side

$$f_p(\boldsymbol{x}) = \frac{1}{\pi^2} + \frac{1}{130d} \sum_{i=1}^{d} \cosh\big(2(x_i - \pi)\big), \qquad \boldsymbol{x} \in [0, 2\pi]^d. \tag{55}$$

However, the Fourier spectral method requires the forcing term to be periodic and to satisfy the zero-mean condition

$$\int_{[0,2\pi]^d} f_p(\boldsymbol{x}) \, \mathrm{d}\boldsymbol{x} = 0. \tag{56}$$

To meet these requirements, we filter the valid Fourier coefficients $\hat{f}_{\boldsymbol{k}}$ and directly set the zero mode $\hat{f}_{\boldsymbol{0}} = 0$. The filtered function is then reconstructed as $f(\boldsymbol{x}) = \sum_{\boldsymbol{k} \in \mathcal{K}} \mathrm{Re}\big(\hat{f}_{\boldsymbol{k}} e^{\mathrm{i}\boldsymbol{k}\cdot\boldsymbol{x}}\big)$, and the corresponding solution is given by $u(\boldsymbol{x}) = \sum_{\boldsymbol{k} \in \mathcal{K}} \mathrm{Re}\big(\hat{u}_{\boldsymbol{k}} e^{\mathrm{i}\boldsymbol{k}\cdot\boldsymbol{x}}\big)$, where $\hat{u}_{\boldsymbol{k}} = -\hat{f}_{\boldsymbol{k}}/\|\boldsymbol{k}\|_2^2$. Since the relative error between $f$ and $f_p$ is less than $10^{-7}$, we can regard $f$ as an accurate periodic approximation of $f_p$.

Furthermore, to avoid an unsolvable solution for PINNs (due to the complexity), we introduce a set of active dimensions $\mathcal{D} \subset \{1, \ldots, d\}$ and redefine

$$f_p(\boldsymbol{x}) = \frac{1}{\pi^2} + \frac{1}{130|\mathcal{D}|} \sum_{i \in \mathcal{D}} \cosh\big(2(x_i - \pi)\big), \qquad \boldsymbol{x} \in [0, 2\pi]^d. \tag{57}$$

Additionally, we conducted a more challenging benchmark by randomly generating the index set $\mathcal{K}$ and the corresponding coefficients $c$. First, we randomly generate the index set $\mathcal{K}$ with a given total number $N_{\text{valid}}$ based on a distribution with higher density over low-frequency components (excluding $\boldsymbol{k} = \boldsymbol{0}$). Then we generate $\hat{f}_{\boldsymbol{k}} = e^{-\|\boldsymbol{k}\|_1} a_{\boldsymbol{k}}$, $a_{\boldsymbol{k}} \sim \mathcal{N}(0,1)$, to guarantee the coefficient decay condition. Then $f(\boldsymbol{x}) = \sum_{\boldsymbol{k} \in \mathcal{K}} \mathrm{Re}\big(\hat{f}_{\boldsymbol{k}} e^{\mathrm{i}\boldsymbol{k}\cdot\boldsymbol{x}}\big)$ and $u(\boldsymbol{x}) = \sum_{\boldsymbol{k} \in \mathcal{K}} \mathrm{Re}\big(\hat{u}_{\boldsymbol{k}} e^{\mathrm{i}\boldsymbol{k}\cdot\boldsymbol{x}}\big)$ where $\hat{u}_{\boldsymbol{k}} = -\hat{f}_{\boldsymbol{k}}/\|\boldsymbol{k}\|_2^2$.

**Heat equations.** For the heat equation considered in Section 4.2, we construct the initial condition as the solution of the Poisson's equation described above with a set of active dimensions $\mathcal{D}$, i.e., $u(\boldsymbol{x}, 0) = \sum_{\boldsymbol{k} \in \mathcal{K}} \mathrm{Re}\big(\hat{u}_{\boldsymbol{k}} e^{\mathrm{i}\boldsymbol{k}\cdot\boldsymbol{x}}\big)$. Then the solution is given by $u(\boldsymbol{x}, t) = \sum_{\boldsymbol{k} \in \mathcal{K}} \mathrm{Re}\big(\hat{u}_{\boldsymbol{k}} e^{-\|\boldsymbol{k}\|_2^2 t} e^{\mathrm{i}\boldsymbol{k}\cdot\boldsymbol{x}}\big)$.

## J.3. Schrödinger Equations

The time-dependent Schrödinger equation (TDSE) governing the quantum dynamics of the system is given by

$$i\hbar \frac{\partial \Psi(\boldsymbol{x}, t)}{\partial t} = \hat{H} \Psi(\boldsymbol{x}, t), \quad \boldsymbol{x} \in [-15\,\text{Å}, 15\,\text{Å}], \ t \in [0, 0.1\,\text{fs}], \tag{58}$$

where $\Psi(\boldsymbol{x}, t)$ denotes the complex-valued wave function of a single quantum particle.

The Hamiltonian operator $\hat{H}$ corresponding to a $d$-dimensional harmonic oscillator potential is defined as

$$\hat{H} = -\frac{\hbar^2}{2m_e}\nabla^2 + 0.1\,k\,\boldsymbol{x}^2, \tag{59}$$

where $m_e$ is the electron mass, $\hbar$ is the reduced Planck constant, and $k$ denotes the stiffness constant of the harmonic potential.

We solve the TDSE numerically using `QMsolve` (De la Fuente & Lamorena, 2024), with $N_t = 3200$ temporal discretization points and $N_x = 100$ spatial grid points. The initial condition is constructed using the same approach in Appendix J.1.2, followed by normalization to ensure unit probability density.

### J.4. PDEs in Physical Space

In physical PDEs, the physical function is typically given, which means that applying spectral methods requires applying the spectral transform as the first step. However, in high-dimensional problems, directly computing the coefficients through such transforms becomes difficult due to CoD.

There are several approaches for computing sparse coefficients of high-dimensional functions using sparse grids. For example, (Piazzola & Tamellini, 2022) identifies key collocation points to compute coefficients, (Gross et al., 2022) computes coefficients on rank-1 lattices, and (Shen & Yu, 2012) employs hyperbolic cross grids. However, these methods introduce non-negligible errors when the computational scale is not large enough. We argue that such transformation-induced errors would lead to an unfair evaluation of SINNs; therefore, in Sections 4.1, 4.2 and 4.5, we use analytic coefficients directly; in Section 4.3 we choose several dimensions to construct the solution; and in Section 4.2, we use separable solutions. Since the aforementioned approaches can produce high-accuracy coefficients when sufficient computational resources are provided, it is reasonable to directly employ analytic coefficients in our experiments.

As for the inverse transforms, the number of required numerical evaluations remains at a reasonable scale in high-dimensional problems. The computation cost of each value via Equation (3) is $\mathcal{O}\left(dN_{\text{valid}}\right)$, where $N_{\text{valid}}$ is the number of coefficients used in the calculation and $d$ is the dimensionality. Validating the accuracy of a model for high-dimensional PDEs on a finite set of points (typically $\sim 10^3$) is a widely recognized assessment methodology in PINNs (Cen & Zou, 2024; Shi et al., 2024; Yu et al., 2026). Since we adopt the same strategy, the inverse transforms in SINNs do not suffer from CoD.

## K. 2D Navier-Stokes Equations with Missed Coefficients

Here we consider the incompressible 2D Navier-Stokes equation:

$$\begin{aligned}
\frac{\partial u}{\partial x} + \frac{\partial v}{\partial y} &= 0, \\
\frac{\partial u}{\partial t} + u\frac{\partial u}{\partial x} + v\frac{\partial u}{\partial y} &= -\frac{1}{\rho}\frac{\partial p}{\partial x} + \nu\left(\frac{\partial^2 u}{\partial x^2} + \frac{\partial^2 u}{\partial y^2}\right), \\
\frac{\partial v}{\partial t} + u\frac{\partial v}{\partial x} + v\frac{\partial v}{\partial y} &= -\frac{1}{\rho}\frac{\partial p}{\partial y} + \nu\left(\frac{\partial^2 v}{\partial x^2} + \frac{\partial^2 v}{\partial y^2}\right).
\end{aligned} \tag{60}$$

We set the density $\rho = 1$, Reynolds number Re $= 100$, computational domain $[0, 2\pi]^2 \times [0, 1]$, and viscosity $\nu = 2\pi/\text{Re}$. We set the index set $\mathcal{K} = [-25, 24]^2 \cap \mathbb{Z}^2$, *i.e.*, $|\mathcal{K}| = 2500$. For the Fourier spectral methods, we use a second-order Adams-Bashforth scheme for temporal discretization with $\Delta t = 10^{-4}$. The boundary condition is periodic because we use a Fourier basis. The initial condition is first generated randomly, after which the solution is evolved using a spectral method up to a fixed time $t = t_0$. The resulting solution at $t = t_0$ is then taken as the initial condition in this numerical experiment, ensuring that the conservation properties of the Navier–Stokes equations are satisfied. We handle the aliasing error by the 2/3 rule (Orszag, 1971b) for both spectral methods and SINN.

The results are demonstrated in Figure 10. If $s < 1$, we mask $(1 - s) \times |\mathcal{K}|$ coefficients based on uniform random sampling. The masked coefficients are the same for both spectral methods and SINN to ensure a fair comparison. The results indicate that, when $s < 1$, SINNs can achieve better accuracy due to their approximation capability even when the PDEs are nonlinear.

## L. Models for High-dimensional Equations

Since the network architecture employed has been discussed above, we only present the loss function here. Suppose $\theta$ is the trainable parameters of the network, $g(\boldsymbol{x})$ is the boundary condition function, $h(\boldsymbol{x})$ is the initial condition function, and $u^\theta$ means $u$ is the output of network. We also provide the training dynamics of PINNs, DRMs and SINNs in Figure 11, Figure 12 and Figure 13 respectively for Poisson's equations.

### L.1. Models for Poisson's Equations

Here we use Equation (50) with periodic boundary conditions for all models.

**PINN.** The PINN uses the residual equation directly, thus the loss function is as follows:

$$\mathcal{L} = \mathcal{L}_r + 100\mathcal{L}_b, \tag{61}$$

where

$$\mathcal{L}_r(\theta) = \frac{1}{N_f} \sum_{j=1}^{N_f} \left| \Delta[u^\theta](\boldsymbol{x}_j^r) - f(\boldsymbol{x}_j^r) \right|^2, \tag{62}$$

$$\mathcal{L}_b(\theta) = \frac{1}{N_b} \sum_{j=1}^{N_b} \left| u^\theta(\boldsymbol{x}_j^b) - g(\boldsymbol{x}_j^b) \right|^2. \tag{63}$$

**DRM.** The DRM uses the energy functional equation of Equation (50) which is as follows:

$$\mathcal{J}[u] = \int_{[0,2\pi]^d} \left( \frac{1}{2} \|\nabla u(\boldsymbol{x})\|_2^2 + f(\boldsymbol{x})u(\boldsymbol{x}) \right) \mathrm{d}\boldsymbol{x}, \tag{64}$$

thus the loss function is as follows:

$$\mathcal{L} = \mathcal{L}_e + 100\mathcal{L}_b, \tag{65}$$

where

$$\mathcal{L}_e(\theta) = \frac{1}{N_e} \sum_{j=1}^{N_e} \left( \frac{1}{2} \left\| \nabla[u^\theta](\boldsymbol{x}_j^e) \right\|_2^2 + f(\boldsymbol{x}_j^e)u^\theta(\boldsymbol{x}_j^e) \right), \tag{66}$$

$$\mathcal{L}_b(\theta) = \frac{1}{N_b} \sum_{j=1}^{N_b} \left| u^\theta(\boldsymbol{x}_j^b) - g(\boldsymbol{x}_j^b) \right|^2. \tag{67}$$

For NDRM, the $\mathcal{L}_e(\theta)$ changes to

$$\mathcal{L}_e(\theta) = \frac{1}{N_e} \sum_{j=1}^{N_e} \left( \frac{1}{2} \left\| \nabla[u^\theta](\boldsymbol{x}_j^e) \right\|_2^2 + \left( u^\theta(\boldsymbol{x}_j^e) - \mathcal{L}_b(\theta) \right) f(\boldsymbol{x}_j^e) \right), \tag{68}$$

**SINN.** The SINN uses the coefficients in the spectral domain, suppose that $\hat{u}_{\boldsymbol{k}}$ is the Fourier coefficients of $u$ and $\hat{f}_{\boldsymbol{k}}$ is the Fourier coefficients of $f$, then the loss function is as follows:

$$\mathcal{L} = \mathcal{L}_f, \tag{69}$$

since $\mathcal{N}$ is a linear operator and Fourier basis has diagonal differential operator, the Equation (9) is simplified to

$$\mathcal{L}_f = \sum_{\boldsymbol{k} \in \mathcal{K}_{N_f}} \left| \|\boldsymbol{k}\|_2^2 \cdot \hat{u}^\theta(\boldsymbol{k}) + \hat{f}_{\boldsymbol{k}} \right|^2. \tag{70}$$

Since SINN holds the periodic boundary condition naturally, the $\mathcal{L}_b$ is removed.

### L.2. Models for Heat Equations

Here we use Equation (53) with periodic boundary conditions for all models.

**PINN.** The PINN uses the residual equation directly, thus the loss function is as follows:

$$\mathcal{L} = \mathcal{L}_r + 100\mathcal{L}_b + 100\mathcal{L}_i, \tag{71}$$

where

$$\mathcal{L}_r(\theta) = \frac{1}{N_f} \sum_{j=1}^{N_f} \left| \Delta[u^\theta](\boldsymbol{x}_j^r, t_j^r) - u_t^\theta(\boldsymbol{x}_j^r, t_j^r) \right|^2, \tag{72}$$

$$\mathcal{L}_b(\theta) = \frac{1}{N_b} \sum_{j=1}^{N_b} \left| u^\theta(\boldsymbol{x}_j^b, t_j^b) - g(\boldsymbol{x}_j^b, t_j^b) \right|^2. \tag{73}$$

$$\mathcal{L}_i(\theta) = \frac{1}{N_i} \sum_{j=1}^{N_i} \left| u^\theta(\boldsymbol{x}_j^i, 0) - h(\boldsymbol{x}_j^i) \right|^2. \tag{74}$$

**DRM.** The DRM uses the energy functional equation of Equation (53) which is as follows:

$$\mathcal{J}[u] = \int_{[0,2\pi]^d} \left( \frac{1}{2} \|\nabla u(\boldsymbol{x}, t)\|_2^2 + u_t(\boldsymbol{x}, t) u(\boldsymbol{x}, t) \right) \mathrm{d}\boldsymbol{x}, \tag{75}$$

thus the loss function is as follows:

$$\mathcal{L} = \mathcal{L}_e + 100\mathcal{L}_b + 100\mathcal{L}_i, \tag{76}$$

where

$$\mathcal{L}_e(\theta) = \frac{1}{N_e} \sum_{j=1}^{N_e} \left( \frac{1}{2} \left\| \nabla[u^\theta](\boldsymbol{x}_j^e, t_j^e) \right\|_2^2 + u_t^\theta(\boldsymbol{x}_j^e, t_j^e) u^\theta(\boldsymbol{x}_j^e, t_j^e) \right), \tag{77}$$

$$\mathcal{L}_b(\theta) = \frac{1}{N_b} \sum_{j=1}^{N_b} \left| u^\theta(\boldsymbol{x}_j^b, t_j^b) - g(\boldsymbol{x}_j^b, t_j^b) \right|^2. \tag{78}$$

$$\mathcal{L}_i(\theta) = \frac{1}{N_i} \sum_{j=1}^{N_i} \left| u^\theta(\boldsymbol{x}_j^i, 0) - h(\boldsymbol{x}_j^i) \right|^2. \tag{79}$$

**SINN.** The SINN uses the coefficients in the spectral domain, suppose that $\hat{u}_{\boldsymbol{k}}$ is the Fourier coefficients of $u$ and $\hat{h}_{\boldsymbol{k}}$ is the Fourier coefficients of $h$, then the loss function is as follows:

$$\mathcal{L} = \mathcal{L}_f + 100\mathcal{L}_i, \tag{80}$$

since $\mathcal{N}$ is a linear operator and Fourier basis has diagonal differential operator, the Equation (9) is simplified to

$$\mathcal{L}_f = \sum_{(\boldsymbol{k}, t) \in \mathcal{K}_{N_f} \times \mathcal{T}_f} \left| \|\boldsymbol{k}\|_2^2 \cdot \hat{u}^\theta(\boldsymbol{k}, t) + \hat{u}_t^\theta(\boldsymbol{k}, t) \right|^2. \tag{81}$$

$$\mathcal{L}_i(\theta) = \sum_{\boldsymbol{k} \in \mathcal{K}_{N_i}} \left| \hat{u}^\theta(\boldsymbol{k}, 0) - \hat{h}_{\boldsymbol{k}} \right|^2. \tag{82}$$

Since SINN holds the periodic boundary condition naturally, the $\mathcal{L}_b$ is removed.

## L.3. Challenging Case

This section gives results for high-dimensional Poisson's equations on the challenging benchmark described in Appendix J.2. The results are summarized in Table 11 for varying dimensionalities and values of $N_{\text{valid}}$. Overall, SINN achieves consistently low numerical errors across all tested dimensions, while maintaining stable performance as the dimensionality increases. In contrast, PINN and DRM exhibit rapidly deteriorating accuracy as either the dimensionality or $N_{\text{valid}}$ increases under the same training configurations. This behavior is consistent with the reported spectral bias phenomena in PINNs (Wang et al., 2022). The effect is particularly pronounced in the $d = 5$ experiments in Table 2. When $N_{\text{valid}} = 2$, the valid frequency set is $|\boldsymbol{k}| = \{3, 9\}$, whereas for $N_{\text{valid}} = 50$, it is $|\boldsymbol{k}| = \{1, 2, 3, 4, 5, 6, 7, 8, 9\}$. Due to the larger spectral gap in the former case, PINN and DRM exhibit significantly larger approximation errors for $N_{\text{valid}} = 2$ than for $N_{\text{valid}} = 50$, despite the latter involving a larger number of valid frequencies. These results suggest that SINN provides a more robust inductive bias for high-dimensional and multiscale problems.

*Table 11.* High-dimensional Poisson's equations with challenging benchmark.

| DIM | $N_{\text{VALID}}$ | PINN (PIRATENETS) | DRM (NDRMS) | SINN |
|---|---|---|---|---|
| 2 | 2 | 5.82E-4 $\pm$ 1.75E-4 | 3.18E-3 $\pm$ 7.93E-4 | 1.51E-5 $\pm$ 1.76E-5 |
|   | 5 | 1.23E-3 $\pm$ 6.89E-4 | 3.40E-3 $\pm$ 3.06E-4 | 7.53E-7 $\pm$ 3.39E-7 |
| 5 | 2 | 7.24E-1 $\pm$ 4.62E-2 | 1.16E-1 $\pm$ 5.51E-3 | 3.40E-7 $\pm$ 3.72E-7 |
|   | 50 | 3.92E-1 $\pm$ 1.65E-2 | 5.69E-2 $\pm$ 1.00E-3 | 6.21E-7 $\pm$ 5.46E-7 |
| 10 | 2 | 5.05E-1 $\pm$ 7.55E-2 | 7.02E-2 $\pm$ 7.51E-4 | 1.87E-7 $\pm$ 1.76E-7 |
|   | 50 | 9.32E-1 $\pm$ 3.25E-2 | 5.38E-1 $\pm$ 1.53E-3 | 2.38E-7 $\pm$ 3.53E-8 |
|   | 100 | > 1 | 9.95E-1 $\pm$ 8.66E-3 | 3.54E-7 $\pm$ 7.39E-8 |
| 30 | 2 | > 1 | > 1 | 4.14E-7 $\pm$ 2.31E-7 |
|   | 150 | > 1 | > 1 | 4.46E-7 $\pm$ 1.65E-8 |
|   | 300 | > 1 | > 1 | 3.36E-7 $\pm$ 1.27E-8 |
| 50 | 2 | > 1 | > 1 | 4.04E-7 $\pm$ 4.78E-8 |
|   | 250 | > 1 | > 1 | 4.92E-7 $\pm$ 2.22E-8 |
|   | 500 | > 1 | > 1 | 2.22E-3 $\pm$ 1.68E-3 |
| 100 | 2 | > 1 | > 1 | 7.53E-7 $\pm$ 1.01E-7 |
|   | 500 | > 1 | > 1 | 2.89E-3 $\pm$ 2.11E-3 |
|   | 1000 | > 1 | > 1 | 6.76E-3 $\pm$ 1.26E-3 |
|   | 10000 | > 1 | > 1 | 4.09E-1 $\pm$ 1.05E-1 |

## L.4. Cost

Here we report the computational cost of these models for high-dimensional Poisson's equations. For PINN, we train with $N_f = 2000$ and $N_b = 2000$ for 50000 iterations. For DRM, we train with $N_e = 50000$ and $N_b = 50000$ for 50000 iterations. For SINN, we train with $N_f = 50000$ for 50000 iterations. Notably, the number of training points for PINN is smaller than for the others due to out-of-memory issues: (i) the network used is larger than the others, and (ii) it requires second derivatives. However, this smaller number does not influence the final results since PINN has already converged before 50000 iterations.

Table 12 shows the model size and training rate of the aforementioned models and establishes that SINN is the most effective approach for high-dimensional PDE solving. Notably, for PINN, due to Fourier feature embeddings, the model size (learnable parameters) does not depend on the number of input channels. Apart from the model size of PINN, the scalability of these methods under high-dimensional conditions varies considerably.

We also report the memory usage and inference time of these models in Table 13. As shown in Table 13, SINN requires substantially less memory than PINN and DRM, primarily because it avoids automatic differentiation of spatial derivatives. In contrast, SINN has higher inference time, since it must process both the real and imaginary parts of spectral coefficients. Nevertheless, the inference cost remains acceptable, and SINN still achieves the fastest training shown in Table 12.

*Table 12.* Cost of high-dimensional Poisson's equations.

| DIM | | PINN | DRM | SINN |
|---|---|---|---|---|
| 2 | MODEL SIZE | 5.27E+5 | 3.09E+4 | 3.28E+4 |
| | RATE | 1.26E+2 | 1.26E+2 | 1.04E+3 |
| 5 | MODEL SIZE | 5.27E+5 | 3.15E+4 | 3.56E+4 |
| | RATE | 4.58E+1 | 1.09E+2 | 5.84E+2 |
| 10 | MODEL SIZE | 5.27E+5 | 3.25E+4 | 3.61E+4 |
| | RATE | 2.10E+1 | 8.93E+1 | 2.27E+2 |
| 30 | MODEL SIZE | 5.27E+5 | 3.65E+4 | 3.81E+4 |
| | RATE | 6.48E+0 | 2.04E+1 | 4.36E+1 |
| 50 | MODEL SIZE | 5.27E+5 | 4.05E+4 | 4.02E+4 |
| | RATE | 3.76E+0 | 1.21E+1 | 2.82E+1 |
| 100 | MODEL SIZE | 5.27E+5 | 5.05E+4 | 4.52E+4 |
| | RATE | 1.46E+0 | 3.45E+0 | 1.46E+1 |

*Table 13.* Memory usage and inference time on high-dimensional Poisson's equations.

| DIM | METRIC | PINN | DRM | SINN |
|---|---|---|---|---|
| 2 | MEMORY USAGE (MB) | 1.26E+3 | 2.27E+3 | 9.95E+2 |
| | INFERENCE TIME (S) | 3.43E-2 | 1.68E-2 | 1.99E-2 |
| 5 | MEMORY USAGE (MB) | 1.27E+3 | 2.27E+3 | 9.95E+2 |
| | INFERENCE TIME (S) | 3.75E-2 | 1.71E-2 | 2.33E-2 |
| 10 | MEMORY USAGE (MB) | 1.78E+3 | 2.27E+3 | 9.95E+2 |
| | INFERENCE TIME (S) | 3.82E-2 | 1.71E-2 | 2.68E-2 |
| 30 | MEMORY USAGE (MB) | 4.86E+3 | 2.28E+3 | 9.95E+2 |
| | INFERENCE TIME (S) | 4.20E-2 | 1.74E-2 | 4.99E-2 |
| 50 | MEMORY USAGE (MB) | 4.87E+3 | 2.29E+3 | 9.95E+2 |
| | INFERENCE TIME (S) | 4.90E-2 | 1.75E-2 | 6.82E-2 |
| 100 | MEMORY USAGE (MB) | 6.95E+3 | 2.31E+3 | 9.97E+2 |
| | INFERENCE TIME (S) | 5.04E-2 | 1.80E-2 | 1.15E-1 |

## M. Metrics

In this paper, we utilize the relative $L^2$ error:

$$\text{RelativeL2} = \frac{\|\boldsymbol{y} - \hat{\boldsymbol{y}}\|_2}{\|\boldsymbol{y}\|_2}, \tag{83}$$

where $\boldsymbol{y} \in \mathbb{R}^N$ is the target value, and $\hat{\boldsymbol{y}} \in \mathbb{R}^N$ is the value predicted by the neural network.

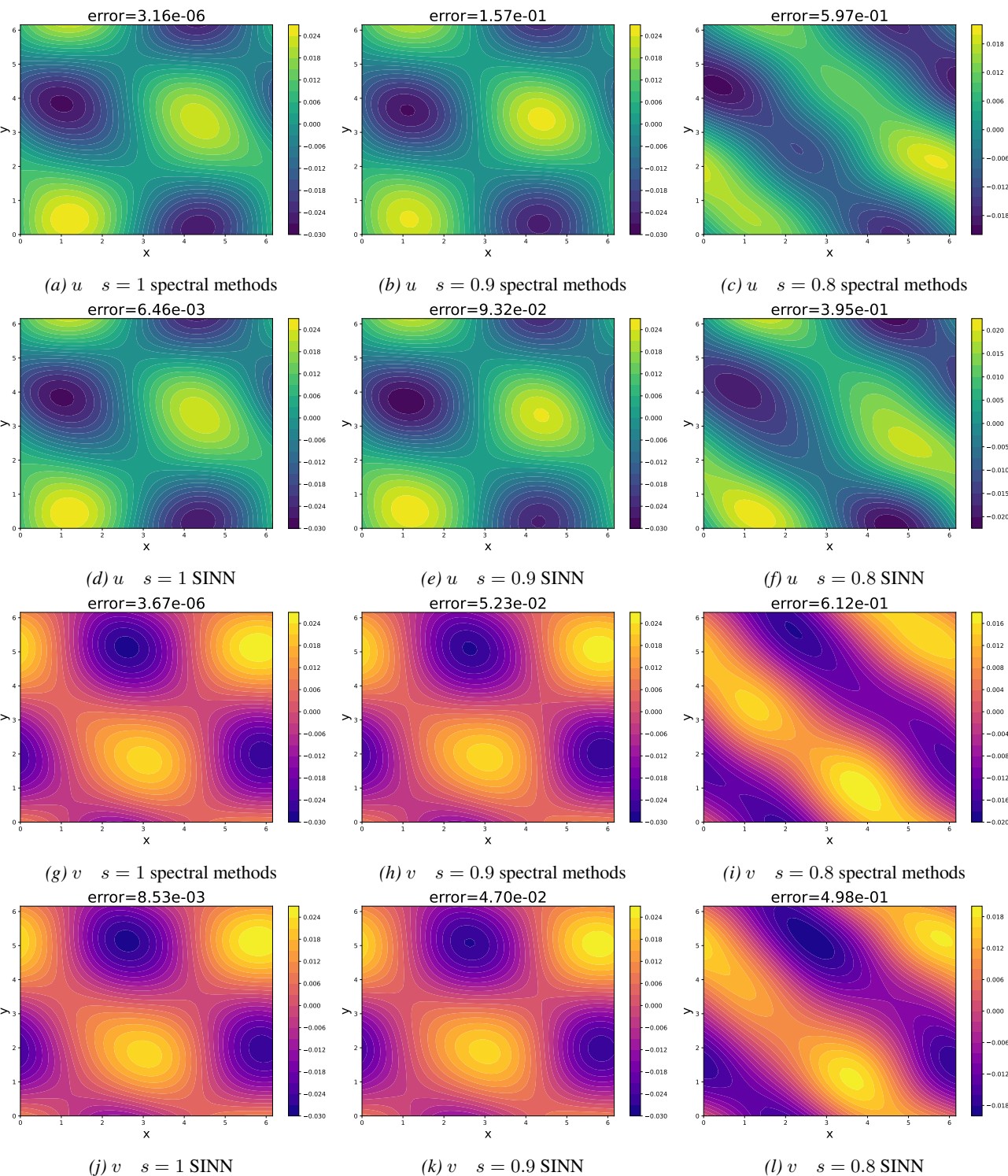

*Figure 10.* The solution $\boldsymbol{u} = (u, v)$ of 2D Navier-Stokes equations. $s = 1$ corresponds to the case where all spectral coefficients are known, $s < 1$ means there are $(1 - s)$ coefficients missing. The top caption of every subfigure is the relative L2 error.

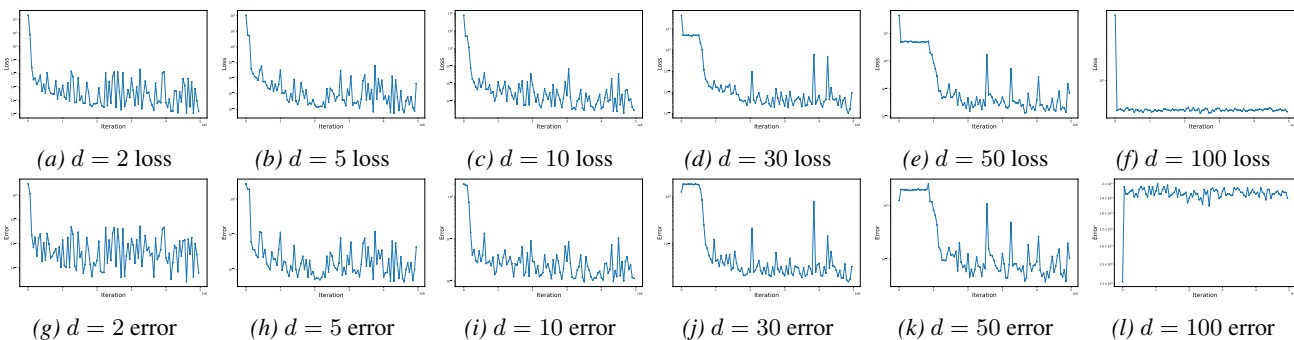

*(a) d = 2* loss    *(b) d = 5* loss    *(c) d = 10* loss    *(d) d = 30* loss    *(e) d = 50* loss    *(f) d = 100* loss

*(g) d = 2* error    *(h) d = 5* error    *(i) d = 10* error    *(j) d = 30* error    *(k) d = 50* error    *(l) d = 100* error

*Figure 11.* Training dynamics of PINN performance for high-dimensional Poisson's equations. The first row reports loss curves and the second row reports relative errors of $u$.

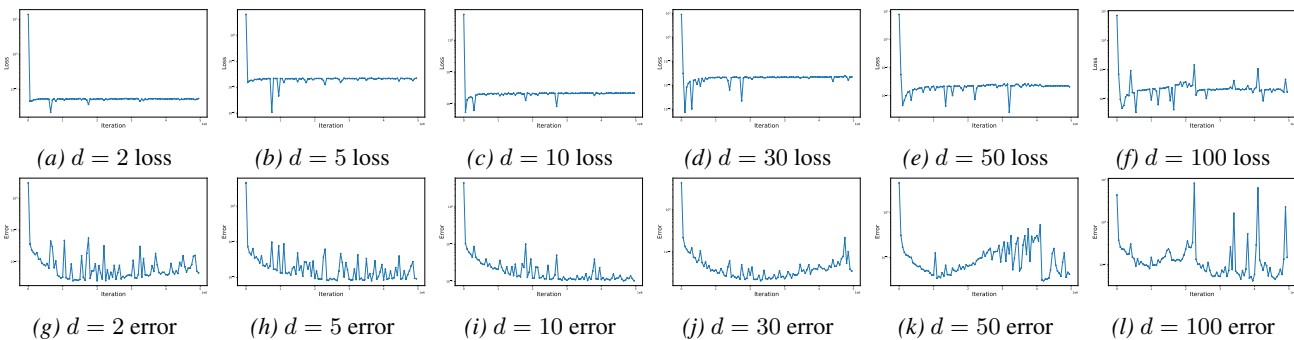

*(a) d = 2* loss    *(b) d = 5* loss    *(c) d = 10* loss    *(d) d = 30* loss    *(e) d = 50* loss    *(f) d = 100* loss

*(g) d = 2* error    *(h) d = 5* error    *(i) d = 10* error    *(j) d = 30* error    *(k) d = 50* error    *(l) d = 100* error

*Figure 12.* Training dynamics of DRM performance for high-dimensional Poisson's equations. The first row reports loss curves and the second row reports relative errors of $u$.

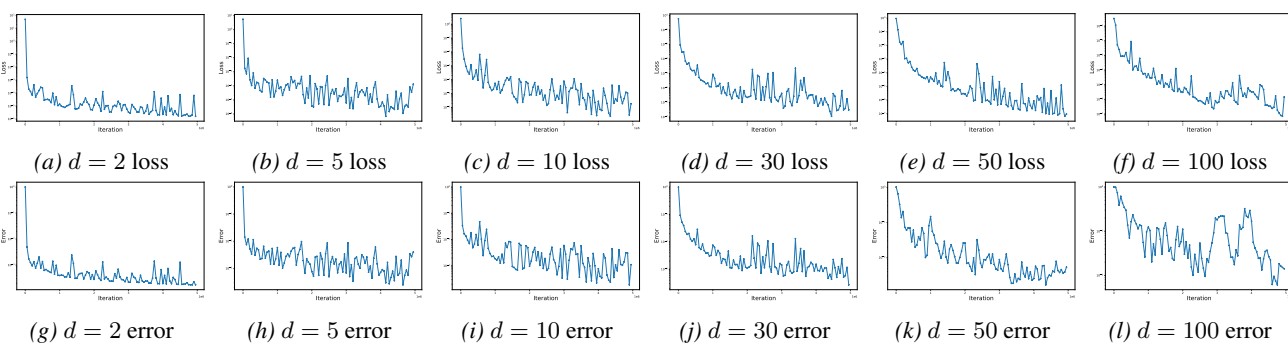

*(a) d = 2* loss    *(b) d = 5* loss    *(c) d = 10* loss    *(d) d = 30* loss    *(e) d = 50* loss    *(f) d = 100* loss

*(g) d = 2* error    *(h) d = 5* error    *(i) d = 10* error    *(j) d = 30* error    *(k) d = 50* error    *(l) d = 100* error

*Figure 13.* Training dynamics of SINN performance for high-dimensional Poisson's equations. The first row reports loss curves and the second row reports relative errors of $u$.

