# OpenReview forum: "Spectral-Informed Neural Networks Outperform Spectral methods in High-dimensional PDEs"
_ICML.cc/2026/Conference — ICML 2026 spotlight_

### Official Review · Reviewer_kTDT · 2026-03-05

**Soundness:** 2
**Presentation:** 1
**Significance:** 2
**Originality:** 3
**Overall Recommendation:** 4
**Confidence:** 4

**Summary:**

This manuscript proposed a modified version of the Spectral-Informed Neural Networks (SINNs) for the efficient solving of high-dimensional PDEs. It integrates the high accuracy of traditional spectral methods with the high-dimensional scalability of neural PDE solvers. The proposed modified SINN architecture consists of the coefficient decay scaling mechanism and the basis embedding module, which infers missing spectral coefficients.

**Compliance With Llm Reviewing Policy:**

Affirmed.

**Key Questions For Authors:**

- N/A. Please refer to the "weaknesses" section above.

**Limitations:**

- Yes.

**Strengths And Weaknesses:**

- **Strengths**
  - Clear motivation to integrate traditional solver's accuracy with neural solver's scalability.
  - Learning spectral coefficients directly as functions of frequency and time is well-grounded in math and is innovative.
- **Weaknesses**
  - Lack of computational efficiency metrics, like training time, inference time, memory consumption, etc. Accuracy competes with efficiency. One metric must be fixed when you conduct experiments on another metric.
  - Lack of training dynamics (convergence plot, etc.). PINNS are known to be difficult to train. Whether SINN converges for other problems or other scenarios remains questionable.
  - Insufficient mathematical rigor. Mathematical analysis, such as convergence guarantees, error bounds relative to spectral truncation, and dependence on dimensionality/sparsity, is all missing.
  - Limited scope. The high-dimensional experiments mainly focus on Poisson equations. It is unclear whether SINN performs well on high-dimensional nonlinear or spatial-temporal PDEs.
  - This approach assumes spectral sparsity or low-dimensional structure.

---

> ### Author Rebuttal · Authors · 2026-03-30
>
> # 1.Lack of computational efficiency metrics,...
> We already report training rate in Appendix K.5, and together with the provided total iterations it gives total training time. We will add inference time and memory usage in Appendix K.5. Partial results are:
> | DIM | Metric | PINN | DRM | SINN |
> | --- | --- | --- | --- | --- |
> | 30 | Memory usage (MB) | 2.60e+3 | 2.34e+3 | 5.36e+2 |
> | 30 | Inference time(s) | 7.62e-2 | 2.64e-2 | 2.47e-1 |
> | 50 | Memory usage(MB) | 4.98e+3 | 2.36e+3 | 1.30e+3 |
> | 50 | Inference time(s) | 7.80e-2 | 2.68e-2 | 4.68e-1 |
>
> As shown in this table, the memory usage of SINN is significantly lower than that of PINN and DRM. This is because SINN avoids automatic differentiation for spatial derivatives. The inference time of SINN is higher than that of PINN and DRM, because SINN needs to handle both the real and imaginary parts of spectral coefficients. However, this inference time remains reasonable and the training time is the lowest.
> # 2.Lack of training dynamics..
> We will plot both loss and relative error. While convergence plots are not included in the current manuscript, the consistently best final errors across different dimensions and PDE settings in experiments indicate stable convergence.
>
> For PINNs, prior [work](https://arxiv.org/abs/2007.14527) shows that spectral bias is the main training difficulty. Since SINNs learn PDEs directly in the spectral domain, this issue is mitigated, and our experiments in Appendix K.4 are consistent with this statement. For SINNs, a more tough challenge is anomalous oscillation, and our modified SINNs are designed to alleviate this problem.
> # 3.Insufficient mathematical rigor ..
> We agree we should add a rigorous analysis in the appendix for high-dimensional PDEs. Below we briefly summarize the analysis idea: In 1D, this [paper](https://www.sciencedirect.com/science/article/abs/pii/S0377042725006922) shows SINNs achieve the same convergence rate as spectral methods. After extending the current analysis to high-dimensional PDEs, we will show, in high dimensional PDEs, the convergence rate is determined by the index set and the regularity of target function like the same in spectral methods. We can give the detailed analysis in the discussion if required.
>
> # 4.Limited scope...
> For nonlinear PDEs, we said in the main text that "generating high-accuracy numerical targets for high-dimensional nonlinear PDEs is difficult". However, we can still discuss the expected behavior theoretically. The main challenge of nonlinear PDEs is the nonlinear term. From a mathematical perspective, nonlinear terms introduce convolution-type interactions in the spectral domain, which may enlarge the effective frequency support (i.e. $\mathcal{K}$). However, in many practical problems (e.g., nonlinear Schrödinger equations), the energy of the solution often remains concentrated in a structured and relatively low-dimensional subset of the spectrum. These problems are easily solved by introducing additional dealiasing operators. In contrast, equations such as the Navier--Stokes equations may exhibit stronger energy transfer toward higher frequencies. Our method has the capability of approximating missing spectral coefficients, and is therefore applicable in such settings.
>
> For high-dimensional spatio-temporal PDEs, we will add experiments on high-dimensional heat equations. Partial results are:
> | DIM | SINN | PINN | DRM |
> | --- | --- | --- | --- |
> | 30 | $1.46e-3 \pm 4.30e-4$ | $3.01e-2 \pm 5.57e-3$ | $3.10e-2 \pm 1.36e-2$ |
> # 5.This approach assumes spectral sparsity...
> As stated in our limitations, high-dimensional problems with genuinely dense coefficients and without low-dimensional structure are currently intractable for all existing methods. For "genuinely dense coefficients", we mean that there is no basis in which the target function admits a sparse representation.
>
> Take PINNs as an example. Let $f(\boldsymbol{x}): \mathbb{R}^d \to \mathbb{R}$ be the target function and let $u_\theta(\boldsymbol{x})$ be the network output. Since the universal approximation theorem shows that MLPs can approximate arbitrary continuous functions, including genuinely dense ones, we consider an MLP where the input of last layer can be denoted as $\{\phi_j\}_{j=1}^w$. Then the output is
>
> $
> u_\theta(\boldsymbol{x}) = \sum_{j=1}^w c_j \phi_j(\boldsymbol{x}),
> $
>
> where $c_j$ are the coefficients in the final linear layer, and the bias can be regarded as the coefficient of $\phi \equiv 1$. Therefore, even a standard neural network represents the target function through a basis expansion. If the target is genuinely dense in the sense above, then either $w$ must be extremely large in high dimensions, or the network fails to approximate the function accurately. Thus, the issue of spectral sparsity or low-dimensional structure is not unique to our method but is a fundamental challenge for all high-dimensional methods, including PINNs and DRMs. We will further clarify this point in the revision.

---

> > ### Author Rebuttal · Reviewer_kTDT · 2026-04-01
> >
> > My concerns have been adequately addressed. I've raised my score.

---

### Official Review · Reviewer_DXUs · 2026-03-07

**Soundness:** 4
**Presentation:** 3
**Significance:** 3
**Originality:** 3
**Overall Recommendation:** 5
**Confidence:** 4

**Summary:**

The paper proposes Modified Spectral-Informed Neural Networks (Modified SINNs) for solving high-dimensional partial differential equations (PDEs). While traditional spectral methods suffer from the curse of dimensionality (CoD) for $d \gg 10$, and standard Physics-Informed Neural Networks (PINNs) often struggle with limited accuracy and efficiency, Modified SINNs operate directly in the spectral domain to compute basis coefficients. The core contribution is the integration of prior knowledge from harmonic analysis into the network architecture via two novel components: a coefficient decay scaler that enforces known physical decay rates (e.g., exponential decay for analytic functions) on high-frequency coefficients, and a basis embedding module that explicitly represents the low-dimensional structure of the spectral basis functions using a coarse collocation point set.

By learning this structure, the network can approximate uncomputed or missing spectral coefficients, bypassing the need to generate full or hyperbolic-cross index sets that become computationally intractable in high dimensions. The authors evaluate their method on middle- and high-dimensional PDEs (including Poisson, heat, convection, and Schrödinger equations). The results demonstrate that Modified SINNs successfully recover masked spectral coefficients, outperform Sparse Grid Spectral Methods (SGSM) when spectral information is incomplete, and achieve superior accuracy compared to baseline PINNs and Deep Ritz Methods (DRMs) in high-dimensional settings. Additionally, the authors propose a practical hybrid SGSM+SINN strategy that leverages exact SGSM computations for unmasked frequencies alongside neural approximations for missing high-frequency components.

**Compliance With Llm Reviewing Policy:**

Affirmed.

**Final Justification:**

The authors have addressed or committed to address my concerns. I believe this is an interesting and original paper representing a solid contribution to the literature. I expect it to be of interest to the SciML community reading and attending ICML. Therefore, I have raised by score to 5, Accept.

**Key Questions For Authors:**

1. The use of explicitly separable functions for the high-dimensional benchmarks (e.g., $f(x) = \prod_{i=1}^d \tilde{f}(x_i)$) is a significant caveat that is only detailed in the appendix. While the computational necessity of doing so to generate exact target coefficients is understandable, it weakens the claim that Modified SINNs can generalize to complex, coupled high-dimensional PDEs. Can you provide an experiment, even in a moderate dimension like $d=5$ or $d=6$ where ground truth coefficients might still be computable without assuming separability, to explicitly demonstrate that the network successfully learns the coupled integration via the position embedding mentioned in Remark F.1?
2. In Section 4.1, the network is evaluated on its ability to approximate uniformly masked coefficients. The manuscript acknowledges this is unrealistic since significant coefficients concentrate at low frequencies. Why was this unrealistic distribution used for the primary capability demonstration in Table 1 and Figure 2? Could you provide the results for these specific experiments using the realistic, low-frequency-biased masking distribution that was introduced later in Section 4.4?
3. The exclusion of PINNs and DRMs from the middle-dimensional comparisons in Section 4.2 is justified by stating they generally perform worse than spectral methods. However, including them would provide a highly valuable empirical picture of the exact crossover point where Sparse Grid Spectral Methods (SGSM) begin to fail due to the curse of dimensionality and neural methods take over. Would it be possible to include PINN/DRM baselines in Figure 3 to illustrate this transition?
5. The core mathematical limitations and the construction of the benchmark datasets are heavily relegated to the appendices . To improve the manuscript's transparency, would you consider moving the explicit definitions of the target signals (particularly the separability assumption) into the main text of the experimental section?

If the authors address the concerns in this section and the rest of the review, I would be open to changing my recommendation in the discussion phase to "Accept".

**Limitations:**

The authors have partially, but not completely, discussed the limitations of their work. They should be commended for dedicating a specific section to "Limitations and future research". They are upfront about the fact that their methodology fundamentally relies on the assumption of sparse coefficients. They candidly acknowledge that if a high-dimensional problem has genuinely dense coefficients and lacks an underlying low-dimensional structure, it remains intractable for this method (and all current methods). Furthermore, they honestly note that Modified SINNs can fail in extreme masking scenarios, such as when too many critical low-frequency coefficients are hidden. The authors also include a standard impact statement noting no specific negative societal consequences, which is appropriate for a theoretical PDE solver.

While the authors were transparent about the mathematical limits of the model, the same does not apply to the limitations of their empirical validation. As noted in the weaknesses, the authors rely heavily on explicitly separable functions to generate the ground-truth target coefficients for their high-dimensional experiments. This is a massive constraint on the method's practical applicability, yet it is entirely absent from the main text's limitations section and only mentioned in the appendices. To adequately discuss their limitations, the authors should bring this separability caveat into Section 5 so readers clearly understand the boundaries of the high-dimensional claims.

**Strengths And Weaknesses:**

## Strengths

* The authors successfully bypass the notorious inefficiencies of using neural networks as spatial mesh-free solvers. By operating strictly in the spectral domain, the network avoids the computationally expensive calculation of spatial derivatives via automatic differentiation. Instead of hoping a black-box MLP learns the physics, the architecture explicitly enforces the bounds of harmonic analysis. The coefficient decay scaler forces the network to respect the known algebraic or exponential decay rates of smooth and analytic functions. Combined with the basis embedding module, this makes the optimization highly constrained and theoretically sound.
* The paper accurately identifies that even traditional "tricks" for scaling spectral methods, such as hyperbolic cross index sets, ultimately succumb to the curse of dimensionality for $d > 10$ due to exponential memory costs. Using a neural network to approximate these uncomputable high-dimensional coefficients is a highly justified use case.
* The proposed SGSM+SINN hybrid approach is highly pragmatic. By relying on exact Sparse Grid Spectral Methods to compute the unmasked low frequencies and deploying the neural network purely to predict the masked/missing high frequencies, the method reduces errors by up to several orders of magnitude.

## Weaknesses

* The paper makes strong claims about solving PDEs up to $d=100$. However, it is buried deep in the appendix that the target functions (forcing terms and initial conditions) used for these high-dimensional benchmarks are explicitly constructed to be separable functions. Without separability, computing the exact ground truth coefficients on a grid of size $N^d$ may be computationally impossible. Relying heavily on separable benchmarks severely weakens the claim that this method scales practically to general, non-separable high-dimensional PDEs.
* In the initial coefficient approximation experiments (Section 4.1), the authors evaluate the network using uniformly distributed random masking of coefficients. The authors admit this is unrealistic, as significant coefficients are concentrated at low frequencies. While a more realistic distribution is used later, establishing the core capability on a fundamentally flawed benchmark leaves something to be desired.
* As acknowledged in the limitations, the method is entirely dependent on the underlying PDE yielding a solution with a low-dimensional structure that induces sparsity in the spectral domain. High-dimensional problems with genuinely dense coefficients, shocks, or sharp discontinuities may remain intractable for this method.
* The authors explicitly exclude PINNs and Deep Ritz Methods (DRMs) from the middle-dimensional comparisons against SGSM, claiming they generally perform worse. While likely true, omitting these baselines denies the reader a complete empirical picture of where the crossover in performance actually occurs. I recommend to add these baselines if possible.
* The manuscript is somewhat disorganized and hard to follow. It dedicates excessive space to detailing hyperbolic cross index sets only to conclude they are insufficient, while critical methodological limitations, such as the reliance on separable benchmark functions and the true definition of the test signals, are relegated to the appendices. In addition, I feel that some of the mathematical notation can be simplified and cleaned up to help readability. I recommend to avoid using multiple similar symbols and to avoid subscripting and superscripting where not strictly necessary, among other improvements. It is difficult to keep track of what each subscript or superscript represents and sometimes omitting things that are unambiguous, or only one instance of the relevant object appears in the paper, can improve the manuscript.

---

> ### Author Rebuttal · Authors · 2026-03-30
>
> # Questions
> ## 1.Can you provide an experiment, even in a moderate dimension ...
> Yes. We can add an experiment at $d=5$ and $d=6$, directly considering the non-separable benchmark from DFVM:
>
> $
> u(\boldsymbol{x})=\left( \sum_{i=1}^d \frac{1}{d} x_i \right)^2 + \sin\left( \sum_{i=1}^d \frac{1}{d} x_i \right), \quad \boldsymbol{x}\in[-\pi,\pi]^d.
> $
>
> This directly tests coupled interactions and allows us to verify whether the position embedding captures cross-dimensional integration. Partial results are:
>
> | DIM | s | Error | Error (Missing) |
> | --- | --- | --- | --- |
> | 5 | 1 | $6.93e-4 \pm 1.47e-4$ |  |
> | 5 | 0.9 | $4.76e-2 \pm 5.13e-2$ | $1.80e-1 \pm 1.92e-1$ |
> | 5 | 0.8 | $8.97e-2 \pm 1.14e-1$ | $2.43e-1 \pm 3.09e-1$ |
> | 6 | 1 | $2.54e-3 \pm 4.25e-4$ |  |
> | 6 | 0.9 | $7.54e-3 \pm 4.48e-3$ | $2.56e-2 \pm 1.55e-2$ |
> | 6 | 0.8 | $2.04e-2 \pm 1.30e-2$ | $5.49e-2 \pm 3.82e-2$ |
> ## 2. In Section 4.1, the network is evaluated...
> When using the low-frequency-biased masking distribution, the missing frequencies become less important, so the gap between Modified SINNs and SGSM decreases considerably. Thus, under this setting for Section 4.1, the results more or less align with those in Section 4.4. However, for Section 4.1, since the goal is to demonstrate the capability of Modified SINNs rather than its practical application, using uniformly masked coefficients produces a larger performance gap and gives a more impressive demonstration of the method's capability. For real problems, as in Section 4.4, low-frequency-biased masking is more appropriate and reasonable since it reflects the actual distribution of spectral coefficients.
>
> We understand that providing the full experiments is valuable. We will add a table in the style of Table 1 for the low-frequency-biased masking setting (with $s=0.8$) in the appendix. Partial results are:
> | Dim | Error | Error (Missing) |
> | --- | --- | --- |
> | 2 | 1.08e-3 | 5.44e-3 |
> | 5 | 3.67e-2 | 1.54e-1 |
> ## 3.The exclusion of PINNs and DRMs...
> Yes, this is a valuable suggestion. We will include PINN and DRM baselines for the middle-dimensional regime. Since Figure 3 is already complex enough, we will report the comparison in an appendix table. Partial results are:
> | Equation | DIM | PINN | DRM |
> | --- | --- | --- | --- |
> | Poisson | 3 | 6.73e-3 | 1.00e-2 |
> | Poisson | 6 | 6.95e-1 | 9.51e-2 |
> | Poisson | 8 | >1 | 2.71e-1 |
> | Heat | 3 | 6.60e-2 | 2.87e-1 |
> | Heat | 6 | >1 | 9.93e-1 |
> ## 4.The core mathematical limitations, ...
> Not all our experiments are separable (e.g., H.1, J, and K.4 use non-separable targets), but we agree that placing these details mainly in the appendix can cause confusion. In the revision, we will move representative non-separable examples to the main experimental section and explicitly label which experiments use separable targets.
> # Weaknesses
> ## 5.The paper makes strong claims...
> We agree with this concern and appreciate the clarification. In the main text, we already note that "Since both the forward and inverse transforms of spectral methods suffer from CoD for high-dimensional functions, we discuss the algorithms and costs of both transforms in Appendix I.4.", with details and reviews of current methods in Appendix I.4. We will make this limitation more explicit in the main paper: our method focuses on learning in spectral space, while obtaining high-accuracy ground-truth coefficients is beyond the scope of this work.
> ## 6. As acknowledged in the limitations ...
> Shocks and sharp discontinuities can induce Gibbs phenomena in spectral approximations; filtering or modified Fourier reconstructions (e.g., $\sigma_k e^{ikx}$) can mitigate but not fully remove this difficulty. However, publicly available high-dimensional discontinuous PDE problems and benchmarks are currently limited due to their complexity.
>
> For the sparsity, as stated in our limitations, high-dimensional problems with genuinely dense coefficients and without low-dimensional structure are currently intractable for all existing methods, including ours. We will clarify this statement and scope more explicitly (also in our response to Reviewer 3 for Question 5).
> ## 7.The manuscript is somewhat....
> Thank you for paying attention to the hyperbolic cross. We have already tried our best to keep this part concise. The story is as follows: if one reads Shen & Yu (2010, 2012), it may seem that these approaches avoid the curse of dimensionality. However, since CoD is still a well-known challenge, there must be some limitations. In discussions with researchers working on spectral methods, these detailed limitations are also not always clear. This is why we included this discussion and considered it one of our contributions: "we reveal that traditional sparse grid spectral methods are still limited by the curse of dimensionality in high-dimensional problems".
>
> Thanks for the recommendation of mathematical part. We will polish our symbols and improve the readability.

---

> > ### Author Rebuttal · Reviewer_DXUs · 2026-03-31
> >
> > The authors have addressed or committed to address my concerns.

---

### Official Review · Reviewer_3ec5 · 2026-03-13

**Soundness:** 3
**Presentation:** 3
**Significance:** 3
**Originality:** 4
**Overall Recommendation:** 5
**Confidence:** 3

**Summary:**

The paper proposes Modified Spectral-Informed Neural Networks (Modified SINNs) to address the numerical solution of high-dimensional partial differential equations (PDEs). Building upon the SINN framework, the authors introduce two modules rooted in harmonic analysis: the Coefficient Decay Scaler and the Basis-Dependent Embedding. This approach avoids the complex computation of spatial derivatives and reduces memory consumption by operating in the spectral domain. Experimental results demonstrate that the model effectively approximates missing spectral coefficients in middle-dimensional problems and significantly outperforms PINNs and DRM in high-dimensional settings in terms of accuracy and efficiency.

**Compliance With Llm Reviewing Policy:**

Affirmed.

**Key Questions For Authors:**

In Appendix G, the paper derives a differentiation matrix M_alpha for sparse coefficients to circumvent the curse of dimensionality. However, in extremely high dimensions (e.g., d=100), what is the engineering overhead for dynamically maintaining, searching, and matching these high-dimensional multi-indices to construct sparse operators? Could this become a practical performance bottleneck during large-scale computations?

**Limitations:**

yes

**Strengths And Weaknesses:**

Strengths:
1. Modified SINNs deeply integrate classical harmonic analysis theory, providing a strong inductive bias for the model. This effectively restricts overestimation in high-frequency regions, leading to more stable model training.
2. The experiments demonstrate exceptional high-dimensional scalability. In tests involving Poisson equations with up to 100 dimensions, the model's accuracy and computational rate significantly lead strong baseline models such as PirateNets and NDRMs.
3. The paper provides a clear error convergence theorem (Theorem 3.1) for the basis embedding module, proving its approximation capability under specified rank conditions.

Weaknesses:
The paper performs excellently on linear problems at d=100. However, for highly nonlinear systems (such as high-dimensional Navier-Stokes equations), it only presents results for d=2. Under strong nonlinearity, the distribution of spectral coefficients becomes extremely complex and exhibits severe mode coupling. It currently lacks sufficient high-dimensional experimental support to verify whether the proposed static decay prior (Decay Scaler) can still effectively capture these dynamical features.

---

> ### Author Rebuttal · Authors · 2026-03-30
>
> # 1.Under strong nonlinearity, the ... whether the proposed static decay prior (Decay Scaler) can still effectively capture these dynamical features.
>
> The decay prior is motivated by the bound $|\hat u_{\boldsymbol{k}}| \le C\,e^{-\|\boldsymbol{\alpha}\cdot \boldsymbol{k}\|_1}$, and $\boldsymbol{\alpha}$ is learnable. For the amplitude factor $C$, it can be regarded as a learnable scaling term, absorbed into the main MLP (with input $(\boldsymbol{k},t)$), so the overall coefficient model is adaptive rather than a fixed static template.
>
> For 3-dimensional Navier--Stokes turbulence equations (since the Navier--Stokes equations describe fluid dynamics, the spatial dimension is at most 3), high-frequency terms can indeed be generated during evolution, i.e., coefficients of high-frequency modes may increase to non-negligible values. This issue affects not only our Decay Scaler, but also classical spectral methods. A standard treatment is the 2/3 rule, which enforces decay on the highest 1/3 modes. This method has been widely used to obtain high-accuracy numerical solutions in spectral methods (e.g., [JHTDB](https://turbulence.idies.jhu.edu/home)). Therefore, under this standard treatment, the Decay Scaler can still capture these dynamical features. For other equations, such as Allen--Cahn and high-dimensional nonlinear Schrödinger-type equations, the nonlinear terms typically show negligible high-frequency growth, so the Decay Scaler can capture the dynamics more efficiently. Thus, regardless of how the nonlinear terms are handled, the decay pattern is consistent with that in our current experiments. Therefore, the current results can already provide empirical support that the Decay Scaler can capture such dynamical features.
>
> # 2.The paper derives a differentiation matrix $M_{alpha}$ ... Could this become a practical performance bottleneck during large-scale computations?
>
> Thank you for paying attention to this point. The construction cost of $M_{\alpha}$ is $\mathcal{O}(N_x^3 + N_x|\mathcal{K}|)$ (Appendix G notation: $N_x$ is the number of Chebyshev--Lobatto nodes per dimension, and $\mathcal{K}$ is the sparse multi-index set). This complexity is not explicitly exponential in $d$; instead, it is controlled by $|\mathcal{K}|$. In other words, regardless of how large $d$ is, the construction remains efficient as long as the spectral index set is sparse. The key reason is that we do not compute the full tensor operator in (41), but directly compute valid sparse index mappings.
>
> However, your concern is right: as $d$ increases, $|\mathcal{K}|$ typically grows. So yes, this can become a real practical bottleneck when $|\mathcal{K}|$ is pretty large, and we will add this discussion, as well as the complexity, explicitly in Appendix G.

---

> > ### Author Rebuttal · Reviewer_3ec5 · 2026-04-02
> >
> > The authors have addressed or committed to address my concerns.

---

### Decision · Program_Chairs · 2026-04-30

**Decision:**

Accept (spotlight)

**Comment:**

Although spectral methods are highly accurate numerical methods for solving PDEs, it is well known that they are inefficient in high-dimensional cases. In this paper, the authors present an improved version of SINNs, which is a method that combines the spectral method with PINNs, by incorporating coefficient decay based on harmonic analysis. The reviewers acknowledge that this method has a strong theoretical foundation. Although several concerns were raised in the initial review, all reviewers agreed that these concerns had been fully resolved after the discussion. Therefore, this paper should be accepted.